# Spin-related Cu-Co pair to increase electrochemical ammonia generation on high-entropy oxides

Shengnan Sun [1,10], Chencheng Dai[2,3,10], Peng Zhao [4,10], Shibo Xi[5], Yi Ren [1], Hui Ru Tan [1], Poh Chong Lim[1], Ming Lin [1], Caozheng Diao [6], Danwei Zhang [1], Chao Wu[5,7], Anke Yu[2], Jie Cheng Jackson Koh[2], Wei Ying Lieu [1,8], Debbie Hwee Leng Seng[1], Libo Sun [3,9], Yuke Li[4], Teck Leong Tan [4], Jia Zhang [4,11] ✉, Zhichuan J. Xu[2,11] ✉ & Zhi Wei Seh [1,11] ✉

The electrochemical conversion of nitrate to ammonia is a way to eliminate nitrate pollutant in water. Cu-Co synergistic effect was found to produce excellent performance in ammonia generation. However, few studies have focused on this effect in high-entropy oxides. Here, we report the spin-related Cu-Co synergistic effect on electrochemical nitrate-to-ammonia conversion using high-entropy oxide $Mg_{0.2}Co_{0.2}Ni_{0.2}Cu_{0.2}Zn_{0.2}O$. In contrast, the Li-incorporated MgCoNiCuZnO exhibits inferior performance. By correlating the electronic structure, we found that the Co spin states are crucial for the Cu-Co synergistic effect for ammonia generation. The Cu-Co pair with a high spin Co in $Mg_{0.2}Co_{0.2}Ni_{0.2}Cu_{0.2}Zn_{0.2}O$ can facilitate ammonia generation, while a low spin Co in Li-incorporated MgCoNiCuZnO decreases the Cu-Co synergistic effect on ammonia generation. These findings offer important insights in employing the synergistic effect and spin states inside for selective catalysis. It also indicates the generality of the magnetic effect in ammonia synthesis between electrocatalysis and thermal catalysis.

Ammonia is a very important fertilizer and an alternative carbon-free hydrogen carrier[1–3]. Currently, ammonia is manufactured from dinitrogen and hydrogen via the energy-intensive Haber-Bosch process at high temperature and high pressure. This makes the green alternative, i.e., dinitrogen[4–6] and nitrate[7] electro-reduction to ammonia, attractive. Compared with dinitrogen which has limited solubility in aqueous electrolytes and a strong N≡N bond, nitrate, as a pollutant in groundwater, is more facile to be reduced due to the high solubility and low N-O dissociation energy[5,8]. However, the process of nitrate reduction is more complex, consisting of the formation of nitrite, nitric

[1]Institute of Materials Research and Engineering (IMRE), Agency for Science, Technology and Research (A*STAR), 2 Fusionopolis Way, Innovis #08-03, Singapore 138634, Republic of Singapore. [2]School of Materials Science and Engineering, Nanyang Technological University, 50 Nanyang Avenue, Singapore 639798, Republic of Singapore. [3]The Cambridge Centre for Advanced Research and Education in Singapore, 1 CREATE Way, Singapore 138602, Republic of Singapore. [4]Institute of High Performance Computing (IHPC), Agency for Science, Technology and Research (A*STAR), 1 Fusionopolis Way, #16-16 Connexis, Singapore 138632, Republic of Singapore. [5]Institute of Sustainability for Chemicals, Energy and Environment (ISCE²), Agency for Science, Technology and Research (A*STAR), 1 Pesek Road, Jurong Island, Singapore 627833, Republic of Singapore. [6]Singapore Synchrotron Light Sources (SSLS), National University of Singapore, 5 Research Link, Singapore 117603, Republic of Singapore. [7]College of Materials Science and Engineering, Sichuan University, Chengdu 610065, China. [8]Pillar of Engineering Product Development, Singapore University of Technology and Design, 8 Somapah Road, Singapore 487372, Republic of Singapore. [9]Department of Chemistry, City University of Hong Kong, Hong Kong SAR, P. R. China. [10]These authors contributed equally: Shengnan Sun, Chencheng Dai, Peng Zhao. [11]These authors jointly supervised this work: Jia Zhang, Zhichuan J. Xu, Zhi Wei Seh. ✉e-mail: zhangj@ihpc.a-star.edu.sg; xuzc@ntu.edu.sg; sehzw@imre.a-star.edu.sg

oxide, nitrous oxide, dinitrogen, and hydrogen, in addition to ammonia[9–12]. The common strategies for high selectivity towards ammonia are to optimize adsorption by alloying and vacancy construction[13,14], impede N-N coupling by dispersing active sites[15] and utilize tandem catalysis by hybridization[16].

Due to their relatively low cost, 3d transition metal-based catalysts are intensively investigated for ammonia generation[17]. Among them, Cu- and Co-based compounds have promising ammonia generation performance. Koper et al. demonstrated that Cu reduces nitrate to nitrite and further to hydroxylamine in alkaline electrolyte[18]. Alloying a second metal such as Zn[19], Sn[20], or Ni[13] is a common approach to modulate the Cu d-band center and the intermediate adsorption energy to increase ammonia generation. Additionally, due to Cu inertness towards hydrogen evolution, it can also be used as a matrix to disperse Ru with low nitrate activation barrier for highly ammonia selectivity and production[21]. Introducing oxygen vacancies in metal oxides like oxygen-rich $CuO_x$[22,23] has also been employed to modulate the electronic state of active sites and adsorption properties, and weaken N-O bonding to promote ammonia generation[14,24]. Moreover, ammonia generation can also be enhanced on the heterojunction by accumulating reactants[10], facilitating the formation of specific intermediates, and suppressing byproducts[25]. For example, the built-in electric field in $TiO_2$/CuCl can be used to accumulate nitrate and lower the NO energy barrier[10]. Zhang's group converted CuO into $Cu/Cu_2O$, making the electron transfer from $Cu_2O$ to Cu for facilitating NOH formation and suppress hydrogen evolution[25].

Co is also a good candidate with decent ammonia selectivity[17]. Indeed, Su et al. developed a $Co_3O_4$/Ti electrode, which is able to produce ammonia more efficiently compared to Ti, Cu, and $Fe_2O_3$/Ti electrodes[26]. Similar to adsorption energy modulation in Cu-based compounds, Huang et al. found that $ZnCo_2O_4$ generates more ammonia compared to $Co_3O_4$, and the charge transfer from Co to Zn produces electron-deficient Co, thereby lowering the energy barrier for $NO_2$ formation and suppressing hydrogen evolution[27]. The synergistic effect of $Co^{2+}$ and $Co^{3+}$ in $Co_3O_4$ on ammonia generation is reported. Specifically, Fu et al. found that $Co^{3+}$ prefers nitrate adsorption and $Co^{2+}$ favors the production of H and this dynamic plays a prominent role in nitrate reduction[28]. However, Wang et al. also indicated that on the O-covered $Co_3O_4$ (311) surface, $Co^{3+}$ interacts more strongly with H than $Co^{2+}$, and this stronger interaction impedes hydrogen evolution[29]. The hybridization of Cu and Co plays a prominent role in ammonia generation. Specifically, ammonia generation is enhanced on the CoO/Cu electrode with an interfacial electric field, in which nitrate adsorption is promoted on the positively charged Cu and nitric oxide adsorption is suppressed on the negatively charged CoO[30]. Such enhancement is also considered due to the synergistic effect from Cu and Co in the recent work[16,31,32]. For example, Schuhmann et al. transformed Cu-Co sulfides into core-shell $Cu/CuO_x$ and Co/CoO phases[16]. The $Cu/CuO_x$ phases preferentially catalyze nitrate reduction to nitrite, which is rapidly reduced to ammonia at the Co/CoO shell[16]. Though obvious Cu-Co synergistic effects have been observed, few studies have focused on the Cu-Co synergistic effect in high-entropy oxides from the perspective of spin states. Spin-related ammonia synthesis from dinitrogen was recently investigated by Chorkendorff and Nørskov et al. using a computational model to explain the electrostatic and spin effects caused by catalytic promoters for enhanced thermal catalysis yield[33]. The spin effect on the intermediates adsorption on the transition metal is also emphasized, for example, high spin state Co has a weaker adsorption ability to the intermediates[33–35].

In this work, we experimentally investigate the spin-related Cu-Co synergistic effect on electrochemical ammonia generation via the high-entropy oxide $Mg_{0.2}Co_{0.2}Ni_{0.2}Cu_{0.2}Zn_{0.2}O$ (RS-20) platform. To the best of our knowledge, this is the first time that high-entropy oxide is reported for electrochemical nitrate reduction to ammonia, and the catalytic performance (Faradaic efficiency, FE 99.3%, yield rate 26.6 mg mg$_{cat}^{-1}$ h$^{-1}$)

is among the best in the literature so far. A multi-component platform disperses the homo-cation for suppressing possible N-N formation and modulates the adsorption energy of the intermediates. $Mg_{0.25}Co_{0.25}Ni_{0.25}Zn_{0.25}O$ (RS-0), $Mg_{0.225}Co_{0.225}Ni_{0.225}Cu_{0.10}Zn_{0.225}O$ (RS-10), $Li_{0.10}Mg_{0.18}Co_{0.18}Ni_{0.18}Cu_{0.18}Zn_{0.18}O$ (Li-RS-18), $Li_{0.20}Mg_{0.16}Co_{0.16}Ni_{0.16}Cu_{0.16}Zn_{0.16}O$ (Li-RS-16) and $Li_{0.30}Mg_{0.14}Co_{0.14}Ni_{0.14}Cu_{0.14}Zn_{0.14}O$ (Li-RS-14) serve as control experiments to demonstrate the significance of Cu in the Cu-Co synergistic effect, as well as the impact of Co spin states on the synergistic effect. We find that the Cu-Co pair with a high spin Co in RS-20 facilitates ammonia generation compared with RS-0 and RS-10. Li incorporation decreases Co spin state and hence hinder the ammonia generation. We also find that the valence states of Co, Ni and Cu in RS-20 and Li-RS-16 decrease slightly after nitrate reduction, and the surface reconstruction can only be observed within a few nanometer-thick from the surface at some localized areas of RS-20, and can hardly be observed on Li-RS-16. We conclude that the coexistence of Cu and Co is crucial for achieving a high spin state in Co needed for improved ammonia generation.

## Results

### Materials and ammonia generation

All the synthesized samples RS-0, RS-10, RS-20, Li-RS-18, Li-RS-16 and Li-RS-14 have a single rock-salt structure (Fm-3m) and no observable impurity (Supplementary Fig. 1), and their bulk information from extended X-ray absorption fine structure (EXAFS), X-ray absorption near edge structure (XANES), O K-edge electron energy loss spectroscopy (EELS) and surface information from Co/Ni/Cu X-ray photoelectron spectroscopy (XPS) can be found in Supplementary Figs. 2–6. All the linear sweep voltammetry (LSV) curves for testing nitrate reduction activity and the electrochemical double-layer capacitance are shown in Supplementary Figs. 7–10 and Fig. 1e. The single rock-salt $Mg_{0.40}Ni_{0.40}Zn_{0.20}O$ (MNZO) and $Mg_{0.25}Ni_{0.25}Cu_{0.25}Zn_{0.25}O$ (MNCZO) were synthesized (Supplementary Fig. 1) and used as references to emphasize the contribution of Cu-Co towards nitrate reduction.

Figure 1a shows the XRD patterns of typical samples RS-0, RS-20 and Li-RS-16. Due to the Jahn-Teller effect of $Cu^{2+}$ (Fig. 1b), RS-20 has a weak (200) diffraction peak relative to other peaks. Cu incorporation increases the intensity ratio I(111)/I(200) from 0.581 (RS-0) to 1.692 (RS-20). This trend is consistent with reports by Berardan et al.[36] Li-RS-16 has a stronger (200) diffraction peak and lower intensity ratio I(111)/I(200) than RS-20, indicating that Li incorporation diminishes the influence from the Jahn-Teller effect. RS-20 also has a relatively broader (200) peak than RS-0 and Li-RS-16, which could be caused by the local disorder induced by the Jahn-Teller effect. The (200) lattice distance of RS-20 can be observed in the TEM image. The oxide contained irregular microparticles (SEM, Fig. 1d).

As shown in Supplementary Fig. 7, the electrochemical nitrate reduction activity increases with the increase of Cu ratio, from RS-0, RS-10 to RS-20, and decreases with the increase of the Li ratio, from RS-20, Li-RS-18, Li-RS-16 to Li-RS-14. For example, as shown in Fig. 1e, a more positive onset potential of the electrolyte with nitrate than that without it can be found on all the catalysts, indicating all the catalysts are active towards nitrate reduction reactions. The activity trend presents an order following RS-20 > Li-RS-16 > RS-0 > MNZO based on both the onset potential and current density in the LSV curves (Fig. 1e). In addition, RS-20 has a higher activity than MNZO and the mixed RS-0 / CuO electrodes, indicating the Cu-Co synergistic effect in RS-20 for nitrate reduction (Supplementary Fig. 10). Considering that the oxides have different electrochemical surface areas, which is proportional to the electrochemical surface capacitances (ECSCs, Fig. 1e inset and Supplementary Fig. 8), the inherent catalyst activities were compared by plotting the current density normalized by the ECSC. Figure 1f shows that the same activity trend is also observed when evaluated by the ECSC-normalized current. Due to the smaller ECSC value of RS-20, the inherent activity of it is

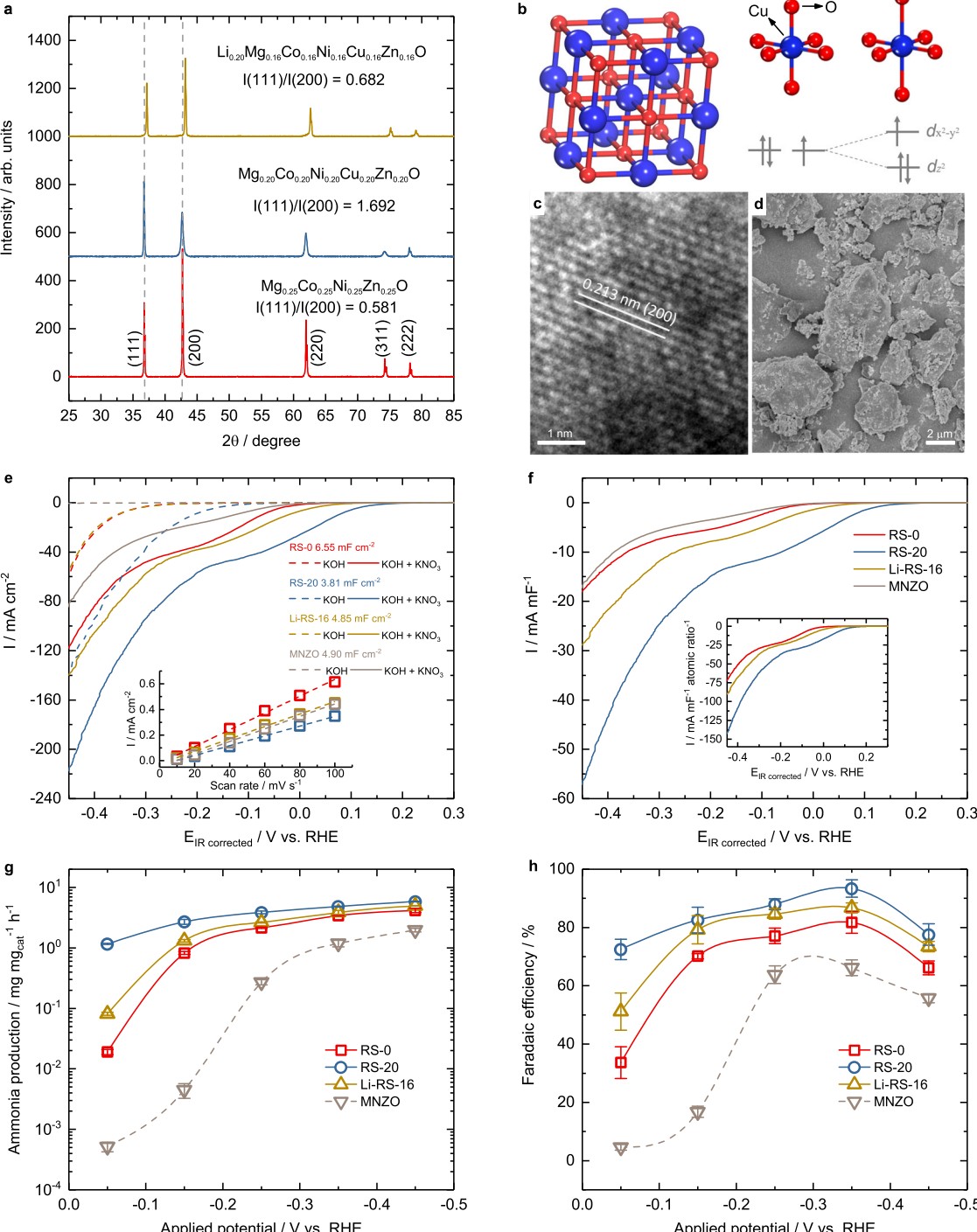

**Fig. 1 | Materials characterization and electrochemical test. a** XRD patterns of RS-0, RS-20, and Li-RS-16. **b** Schematic of the rock-salt structure and Jahn-Teller distortion caused by $Cu^{2+}$. The TEM image (**c**) and SEM image (**d**) of RS-20. **e** LSV curves of nitrate reduction on RS-0, RS-20, and Li-RS-16 and their double-layer capacitance (inset). **f** Capacitance-normalized LSV curves, and atomic ratio (Cu and Co) and capacitance-normalized LSV curves (inset) derived from (**e**). Experimental condition: The mass loading of $1\,mg_{oxide}\,cm^{-2}$ on the glassy carbon substrate (geometry area $0.19625\,cm^2$) for LSV tests in Ar-saturated 1 M KOH with or without $0.1\,M\,KNO_3$ at a scan rate of $10\,mV\,s^{-1}$ with 85% IR correction. The system resistance of the electrochemical cell containing 1 M KOH with and without $0.1\,M\,KNO_3$ is as follow. RS-0, $4.635 \pm 0.493$ Ohm and $4.367 \pm 0.523$ Ohm. RS-20, $6.26 \pm 0.230$ Ohm

and $5.812 \pm 0.138$ Ohm. Li-RS-16, $5.802 \pm 0.135$ Ohm and $4.293 \pm 0.732$ Ohm. MNZO, $5.555 \pm 0.288$ Ohm and $5.135 \pm 0.191$ Ohm. Ammonia production rate (**g**) and FE (**h**) on RS-0, RS-20, Li-RS-16, and MNZO at different applied potentials. The standard deviations come from at least three independent measurements. Experimental condition: The mass loading of $1\,mg_{oxide}\,cm^{-2}$ on the Toray 090 carbon paper ($1.5 \times 1.5\,cm^2$) for nitrate electrolysis at the designed potential for 30 min in a H-cell with FAA-3-PK-130 anion exchange membrane (FuMA-Tech) separating two chambers. Each chamber had 18 mL of 1 M KOH and $0.1\,M\,KNO_3$ electrolyte saturated with Ar. The system resistance of the electrochemical cell containing 1 M KOH and $0.1\,M\,KNO_3$ is as follow. RS-0, $0.861 \pm 0.112$ Ohm. RS-20, $0.828 \pm 0.074$ Ohm. Li-RS-16, $0.802 \pm 0.130$ Ohm. MNZO, $0.896 \pm 0.107$ Ohm.

even more superior than others. The same activity trend remains after normalizing the current density by the total atomic ratios of Cu and Co in the chemical formula (Fig. 1f inset).

To compare the ammonia production rate and FE of these oxides, various potentials were applied from −0.05 to −0.45 V (Fig. 1g and Supplementary Fig. 11) and the ammonia concentration was determined by the UV-vis spectrophotometer (Supplementary Fig. 12). The ammonia production rates show the same trend as the nitrate reduction activity (i.e., RS-20 > Li-RS-16 > RS-0 > MNZO). An obvious gap occurs at −0.05 V. RS-0 shows a higher ammonia production rate than the reference MNZO, indicating that Co is indispensable. Moreover, RS-20 (1.17 mg $mg_{cat}^{-1}$ $h^{-1}$) increases the ammonia production rate by almost 60 times compared to RS-0 (0.019 mg $mg_{cat}^{-1}$ $h^{-1}$); however, incorporating Li decreases ammonia production rate by more than 14 times to 0.081 mg $mg_{cat}^{-1}$ $h^{-1}$. This result indicates that the coexistence of Cu and Co is important for maintaining a high ammonia production rate at a relatively low overpotential, and this ammonia production rate can be deactivated by incorporating Li. The ammonia production rates of RS-0 and Li-RS-16 begin to increase remarkably at −0.15 V and reach close to that of RS-20 at −0.35 V (3.42 mg $mg_{cat}^{-1}$ $h^{-1}$ on RS-0, 4.84 mg $mg_{cat}^{-1}$ $h^{-1}$ on RS-20, 3.88 mg $mg_{cat}^{-1}$ $h^{-1}$ on Li-RS-16), although they remain lower than that of RS-20. The FE order of these oxides is in accordance with their respective ammonia production rate. The FEs of these oxides do not increase monotonically with decreasing applied potential and they peak at −0.35 V and then decrease at −0.45 V. At −0.05 V, RS-0, RS-20 and Li-RS-16 show 33.6%, 72.5% and 51.2% in FE, respectively. At −0.35 V, the FEs of RS-0, RS-20, and Li-RS-16 are 81.8%, 93.4% and 86.7%, respectively. This is due to the competition between ammonia production and the parasitic HER. The obvious HER in the absence of nitrate can be observed, particularly on RS-20. When the applied potential decreases to more negative than the HER thermodynamic potential, the HER is more favored due to its larger charge transfer coefficient for HER than that of *NO hydrogenation. As this disfavors ammonia production, the ammonia FE drops dramatically, particularly observed in those transition metals with a high HER activity, such as Ni. This phenomenon is also reported in other reports[17,21,37]. Therefore, our results on nitrate reduction and ammonia generation demonstrate that RS-20 (4.84 mg $mg_{cat}^{-1}$ $h^{-1}$, FE 93.4%, at −0.35 V) exhibits excellent performance in converting nitrate to ammonia. For the ammonia generation contribution from Ni, Mg and Zn, the pure Ni catalyst is reported to be almost inactive for nitrate reduction in alkaline[13] where nitrogen is the main product of nitrate reduction on Ni-platinized electrode[38]. As $Mg^{2+}$ ($2s^22p^6$) and $Zn^{2+}$ ($3s^23d^{10}$) have fully occupied orbitals, their contributions were limited. Because these oxides have medium/high entropies, similar cation species like Ni and even intercalated $Mg^{2+}$ and $Zn^{2+}$ are well dispersed, which promotes single-atom catalyst (SAC)-like configurations, impeding dinitrogen formation and hydrogen evolution, thus enhancing ammonia selectivity[11]. It is reported that the electrochemical reduction of NO intermediates can lead to $NH_3$ and $N_2$[15,39]. Therefore, to facilitate the generation of ammonia instead of nitrogen, inhibiting N-N coupling on the catalyst surface is necessary. The high-entropy oxides have dispersed or isolated homo-cations, which could play a similar role in inhibiting the N-N coupling, similar to SACs. SACs usually possess high selectivity toward specific products and maximum atom utilization efficiency and quantum size effects[40]. On SACs, the transition metal atoms are dispersed, which minimizes possibility of the N-N coupling from two mono-nitrogen moieties on adjacent sites, which eventually inhibits $N_2$ formation and in turn promotes the $NH_3$ selectivity[15,39]. Compared to SACs, the high-entropy oxides have abundant catalytic sites and high stability, which is beneficial to overcome scaling relation limitations of adsorption energies. They are also treated as potential next-generation catalysts[41]. Our design takes into consideration the poor binding of H to Cu, which contributes to the selective ammonia formation[12].

## Electronic state

Although Cu incorporation seems to increase ammonia production rate due to the synergistic effect between Cu and Co, the mechanism behind Li-induced deactivation remains unclear. As catalytic performance is highly related to the intrinsic properties of catalysts[42], it is necessary to investigate changes in local structure and valence state. The EXAFS spectra (Supplementary Fig. 2) demonstrate similar structure for all the synthesized high-entropy samples. The only exceptions are the shorter Co-O bond distance for samples with Li-incorporation. As displayed in EXAFS spectra (Fig. 2a), RS-0 and RS-20 have the Co-O peak positions at 1.72 Å and 1.69 Å, respectively, while the Co-O peak position in Li-RS-16 decreases to 1.47 Å. It indicates that the Li incorporation causes the dramatic decrease in Co-O length. Contrasting with the Co-O length, small decreases in Ni-O, Cu-O, and Zn-O bond lengths are observed in Li-RS-16 (Figs. 2b, 2c and Supplementary Fig. 2d). Li incorporation changes metal-metal peak positions slightly and weakens the Co/Ni/Zn-metal peak intensity (Figs. 2a, 2b, Supplementary Fig. 2d), but intensifies the Cu-metal peak (Fig. 2c). The weaker Co/Ni/Zn-metal peaks can be attributed to the substitution of a lighter element Li[43]. The stronger Cu-metal peak suggests that Li can diminish the local structure disorder caused by the Jahn-Teller effect on Cu.

To measure the bulk valence state, the normalized μχ value 0.5 in XANES are used, with a higher energy indicating a higher valence state. The Co valence states are estimated from the extrapolation of linear fitting the valence and energy at μχ value 0.5 of CoO and $Co_3O_4$ and these values can be found in Supplementary Fig. 4. The Co valence state does not change obviously with the increasing Cu ratio, but it increases obviously after Li incorporation. RS-0 and RS-20 have similar Co valence states with commercial CoO while Li-RS-16 has a higher Co valence state (-2.39, Fig. 2d and Supplementary Fig. 4). This result is not consistent with that reported by Osenciat et al.[44] Based on TGA and XPS results on $Li_x[MgCoNiCuZn]_{1-x}O$, they reported that when the Li fraction is lower than 1/6, charge compensation occurs via the transition from $Co^{2+}$ to $Co^{3+}$, and when the Li fraction exceeds -0.21, oxygen vacancies form[44]. Because the Co valence state in Li-RS-16 is insufficient to compensate for the change of charge caused by incorporating monovalent Li, it means that other transition metals or oxygen could be oxidized. The energy shift after Li incorporation from Ni K-edge XANES can hardly be observed (Supplementary Fig. 3b). Additionally, although the slight energy shift in Zn K-edge after Li incorporation can be observed (Supplementary Fig. 3d), the increment in Zn valence state cannot be concluded due to the stability of $Zn^{2+}$. O K-edge EELS was used to probe the oxidation state of oxygen[45]. Figure 2f shows that Li-RS-16 has a very prominent oxygen pre-edge at 529.3 eV compared with RS-0 and RS-20. This emergent pre-edge in Li-RS-16 can be attributed to holes in the oxygen orbitals, suggesting that the oxygen could be oxidized[45]. X-ray photoelectron spectroscopy (XPS) results showed no obvious change in the O 1s profile (Fig. 2g). In RS-20, for example, the peaks at 529.7 eV, 531.3 eV, and 532.7 eV are attributed to the oxygen atoms bound to metals, defect sites with low oxygen coordination, and hydroxyl groups or surface-adsorbed oxygen[46]. Though oxygen vacancy is observed, the difference between these oxides is not significant. In addition, no obvious change is observed in the XPS results of Zn and Mg (Supplementary Fig. 13). Thus, Li incorporation causes the coexistence of $Co^{2+δ+}$, $Ni^{2+δ+}$, oxygen with holes, and oxygen vacancy.

The decrease in Co-O bond length (Fig. 2a) and the increase in Co valence state (Fig. 2d) in Li-RS-16 could decrease the Co spin state (i.e. $Co^{3+}$ radius 0.525 Å, low spin; $Co^{2+}$ radius 0.735 Å, high spin)[47]. Co 2p XPS results confirm this assumption (Fig. 2e). An obvious change can be observed from the Co $2p_{3/2}$ main peak. Specifically, RS-0 has two peaks at 780.2 eV and 785.0 eV in the Co $2p_{3/2}$ spectra, which can be assigned to $Co^{2+}$ and the satellite peak[48]. Compared with RS-0, RS-20 has another valence state fraction at -782.2 eV, which could be due to

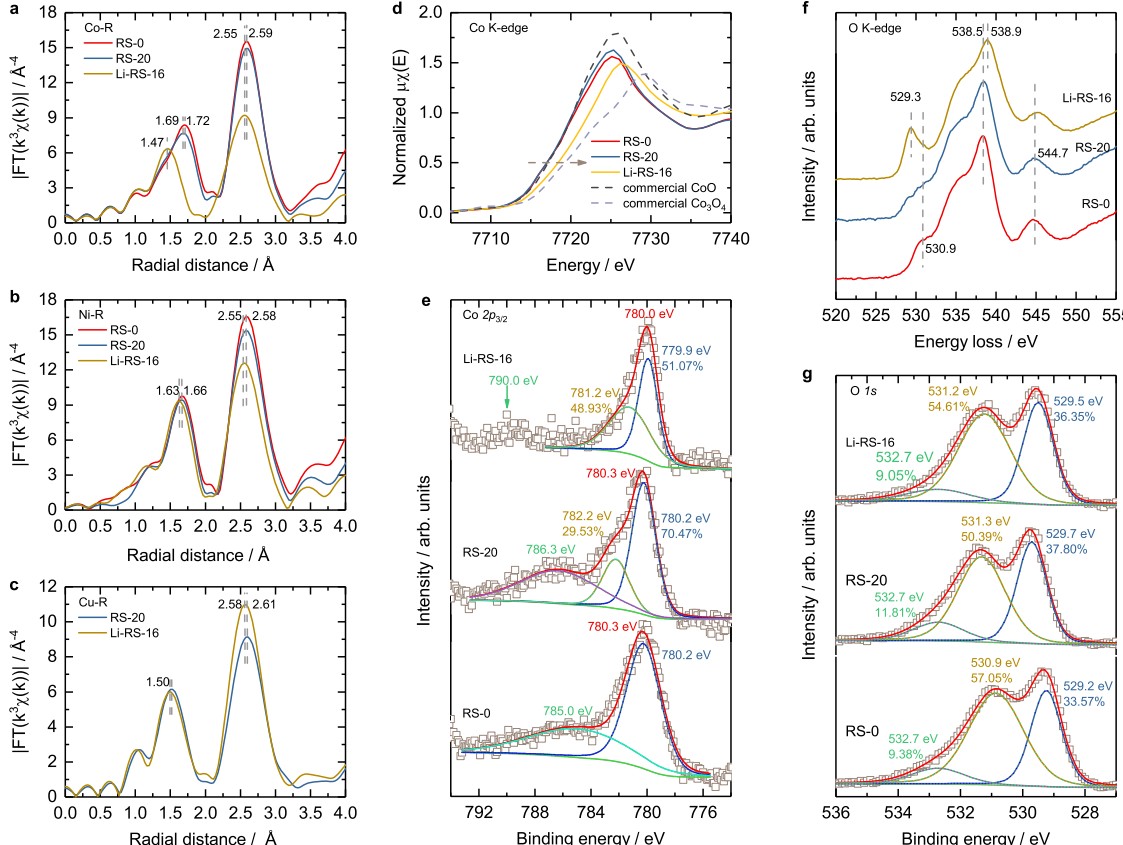

**Fig. 2 | The EXAFS, XANES, XPS and EELS characterization for comparing the local structure, valence state and spin state of Co, Ni, Cu and O in RS-0, RS-20 and Li-RS-16.** EXAFS of (**a**) Co, (**b**) Ni, and (**c**) Cu. **d** XANES of Co with the reference of commercial CoO and Co₃O₄. **e** XPS of Co $2p_{3/2}$ results. **f** O K-edge EELS. **g** XPS of O

1$s$ results. The peak positions are marked by the vertical dash lines and the corresponding labels in panels. The square symbols and solid lines in (**e**) and (**g**) represent the raw data and fitted curves, respectively.

the appearance of $Co^{3+}$. The Cu-induced appearance of $Co^{3+}$ has been reported by Meisenheimer et al.[49] The relative binding energy of $Co^{2+}$ and $Co^{3+}$ is under debate; specifically, the satellite peak is considered a more efficient tool to access the cobalt valence state than the binding energy of the main peak itself[50]. In Li-RS-16, $Co^{2+}$ and a peak at ~781.2 eV can be observed, which could be ascribed to a higher valence state of Co, combining the XANES results. More importantly, the $Co^{2+}$ satellite peak disappears and a weak hump appears at ~790 eV, confirming the existence of $Co^{3+}$[50]. For the spin state, high spin $Co^{2+}$ normally exhibits a rather strong satellite at about 5 eV higher in binding energy at the main peak, while low spin $Co^{2+}$ shows a satellite structure that is less intense and less well resolved[51,52]. Because Li-RS-16 shows no high spin $Co^{2+}$ satellite peak but merely a weak hump corresponding to $Co^{3+}$, we consider it as the lowest spin state among these samples. Such disappearance of satellite peaks is also observed in Li-RS-14, and only trace satellite peak signal can be observed in Li-RS-18 (Supplementary Fig. 5). This spin change caused by Li incorporation is also supported by the vibrating-sample magnetometry (VSM, Supplementary Fig. 14) and soft X-ray test on Co/Ni/Cu L-edges and O K-edge (Supplementary Fig. 15). More discussion can be found in the description of Supplementary Figs. 14, 15. Thus, Li incorporation changes the Co spin state from high spin (RS-20) to low spin (Li-RS-16). This change in the Co spin state could cause the decrease in ammonia generation in Li-RS-16 compared with that in RS-20.

## Materials after nitrate reduction

Another important issue is whether surface reconstruction occurs after nitrate reduction to confirm the stability of the catalyst and the origin of the catalytic activity. The surface change of RS-20 and Li-RS-

16 at −0.35 V for 30 min were first observed by TEM. From the HRTEM images of RS-20 after nitrate reduction (Supplementary Fig. 16a–c), the obvious lattices can be observed at different positions and no reconstruction occurs at the surface. The TEM-EDX mapping does not indicate obvious change in the element distribution (Supplementary Fig. 16d). However, EDX line scans show the higher concentration of Co, Ni and Cu at the surface (Supplementary Fig. 17a), compared with RS-20 prior to nitrate reduction (Supplementary Fig. 19), which means the reconstruction occurs at the surface but only a few nanometers. This surface reconstruction is not uniform, and it only occur at certain localized positions. There is a higher concentration of Mg, Co, Ni Cu and Zn in Supplementary Fig. 17b, while there is no obvious overall element concentration change in Supplementary Fig. 17c. It suggests that the surface reconstruction indeed occurs during nitrate reduction, but only occurs at the very localized areas at surfaces, and the reconstruction layer is very thin. Thus, it can be explained why the HRTEM images and mapping do not indicate the change. This result is different from the previous reports on the obvious reduction of $CuO_x$ during nitrate reduction[14,53–55]. It suggests that more negative potential is needed to reduce the high-entropy oxide $Mg_{0.20}Co_{0.20}Ni_{0.20}Cu_{0.20}Zn_{0.20}O$ during nitrate reduction. In Li-RS-16 after nitrate reduction, HRTEM and TEM-EDX mapping show good crystallization and element distribution around the surface (Supplementary Fig. 20). Meanwhile, the EDX line scan did not indicate such element distribution changes at the surface of Li-RS-16 after nitrate reduction (Supplementary Figs. 21, 22), compared with Li-RS-16 prior to nitrate reduction (Supplementary Fig. 24). Thus, during nitrate reduction, the surface reconstruction is restricted at few nanometers from the surface of

RS-20, and cannot occur at other oxide surfaces. We cannot conclude that the local surface reconstruction determines the high ammonia generation performance in RS-20 yet, while Li incorporation can be proved to inhibit the surface reconstruction.

The valence state and structure after nitrate reduction are further revealed by XANES and EXAFS spectra (Fig. 3). XANES results show that the Co/Ni/Cu K-edges shift to the lower energy direction, indicating the decrease of the Co, Cu and Ni valence states after nitrate reduction. The decrease of EXAFS peak intensity after nitrate reduction demonstrates that the structure becomes less ordered. For Co, according to the fitting of valence states and energies in Supplementary Fig. 4, the valence states of Co in RS-20 and Li-RS-16 decreases from 1.94 to 1.67 and 2.39 to 2.14, respectively. In Fig. 3b, no metallic Co is observed by comparing the Co EXAFS results between our catalysts after nitrate reduction and Co foil. When similar analyses are applied to Ni and Cu, the slight reduction can be observed, and no metallic Ni/Cu signal can be found (Fig. 3c-f). In addition, the $Cu_2O$ pre-edge at 8983.7 eV in XANES is not observed after nitrate reduction (Fig. 3e). It indicates that even though $Cu_2O$ and Cu are formed after nitrate reduction, the ratio was very low. Thereafter, we used EELS to probe the electronic states of Co and Ni and Cu at different locations of the bulk and surface before and after nitrate reduction (Supplementary Figs. 25-28). We found that the Co on the surface is easier to be reduced than that in bulk after nitrate reduction (Supplementary Fig. 29), and the reduction of Cu and Ni is hardly observed. These results are in consistent with the XANES results that the reduction of Cu and Ni is slight. More detailed discussion can be found in the description of Supplementary Fig. 29. We also discuss the stability from the aspects of Gibbs free energy (Supplementary Table 1), standard reduction potential (Supplementary Table 2) and Pourbaix diagrams, and list some cases of the reduction of Cu compounds during or after nitrate reduction (Supplementary Table 3).

## Isotopic labeling and durability

Isotopic labeling using $^1H$ NMR confirmed that the ammonia was produced from nitrate electroreduction on RS-20 rather than contamination (Fig. 4a). The samples were prepared by applying fixed potentials for 2 h. The chemical shift at 6.3 ppm was used as the internal standard[56]. Double peaks appear at the chemical shift at 7.09 and 6.94/6.95 ppm when using $^{15}NO_3^-$ as the reactant at −0.35 and −0.25 V, while triple peaks appear at 7.12, 7.02, and 6.91 ppm when using $^{14}NO_3^-$ as the reactant[10,15]. The ammonia produced under −0.25 V and −0.35 V are consistent comparing the test results by UV-vis spectrophotometry and NMR measurement (Fig. 4b). The nitrate concentration response on RS-20 was also evaluated from 0.01, 0.05, 0.1 to 0.5 M at −0.35 V (Fig. 4c). The results show that ammonia production rate increases from 0.59, 2.77, 5.88 to 14.68 mg $mg_{cat}^{-1}\,h^{-1}$ with the increasing nitrate concentrations. FEs increase from 34.8%, 82.2% to 93.4% when nitrate concentration increases from 10, 50 to 100 mM, respectively, and remain at 93.4% at 0.5 M. This suggests that the higher nitrate concentration can significantly facilitate the ammonia production rate in this system, while the positive effect on FE is only obvious when the nitrate concentration is no higher than 0.1 M. The durability of RS-20 in nitrate reduction was evaluated by 20 consecutive electrolysis in a H-cell reactor under −0.35 V (the best FE condition, Fig. 4d). All ammonia production and FEs remain relatively stable, above 5.65 mg $mg_{cat}^{-1}\,h^{-1}$ and at least 85.18%, which suggests the excellent stability of our catalyst. It is worth noting that the ammonia production rate (5.65 mg $mg_{cat}^{-1}\,h^{-1}$) in the durability test is higher than 4.84 mg $mg_{cat}^{-1}\,h^{-1}$ in the previously mentioned potential-determined ammonia production rate in Fig. 1, while the FE 85.18% is lower than that in potential-determined FE (93.4%). This suggests that the pre-conditioning of the electrode is important for ammonia generation. Our ammonia generation performance is comparable to other works (Supplementary Table 4) in the alkaline solution.

## Flow cell test

Due to the mass transport limitation of the reactant supply and product dissipation, the most recent work by Tarpeh et al. has demonstrated that under a mass transport limitation with a sufficiently negative potential, they enhanced the mass transport by increasing the flow rate (from 10 to 100 mL min$^{-1}$) in a flow cell and thus promote the nitrate reduction activity but lower the ammonia selectivity[57]. In our work, to further improve the ammonia production performance, the flow cell was also employed (Fig. 5a, electrolyte flow rate 5 mL min$^{-1}$, more information can be found in Methods-Flow cell test). Figures 5b, c compare the ammonia production performance in the H-cell and the flow cell with 1 M KOH and 0.1 M $KNO_3$ using RS-20 catalyst. The ammonia production increases with the increasing overpotential, showing the same trend as that in H-cell. Meanwhile, the flow cell achieves a higher ammonia production rate (10.06 mg $mg_{cat}^{-1}\,h^{-1}$ at −0.4 V, FE 66.4%) than the H-cell (Fig. 5b) as a result of the enhanced mass transfer. Different from the trend of ammonia yield rate, the FEs increase first and then decrease with the increasing overpotential. The peak value appears at a more positive potential of −0.2 V, and the peak ammonia FE and yield rate values (99.3%, 5.05 mg $mg_{cat}^{-1}\,h^{-1}$) are higher in the flow cell than that in the H-cell (Fig. 5c). The promotion of both the ammonia selectivity and ammonia production rate is likely due to the enhanced mass transport of both nitrate ions and ammonia products. The more positive peak potential can be attributed to the fact that the HER is also facilitated in the flow cell. In addition, the improvement in ammonia selectivity compared with Tarpeh et al.'s work could be attributed to the strong binding of nitrite on Co[17], low flow rate and less negative potential[57]. Figures 5d, e compare the performance in 1 M KOH with $KNO_3$ of various concentrations. As shown in Fig. 5d, when the nitrate concentration is below 0.1 M, the ammonia yield rate reaches a maximum value when the applied potential is −0.1 V. It is worth mentioning that the FE at −0.1 V for $NO_3RR$ with nitrate concentration of only 10 mM can reach 92.8% in the flow cell, which is significantly higher than the value in the H-cell. This again is a result of the enhanced mass transport brought by the convection in the flow cell. Conversely, in the stagnant H-cell, the concentration gradient in the diffusion layer, which is proportional to the current density, is limited by the small concentration difference between the bulk and surface, and the much thicker diffusion layer thickness. Since the ammonia yield rate is limited with lower nitrate concentration and the HER rate increases with the decreasing applied potential, the FE will inevitably drop. The situation differs when the nitrate concentration is 0.5 M. It can be found in Fig. 5e that the FE with 0.5 M $KNO_3$ is smaller than that of 0.1 M $KNO_3$ when the potential is more positive than −0.2 V. This could be led by obstructed mass transport due to the increased viscosity of higher concentration $KNO_3$ electrolyte. Moreover, there is limited FE drop when the applied potential is more and more negative. The reason might be that the surface is occupied by the abundant adsorbed nitrate reduction reactant and intermediates, so that the HER is hindered and therefore the FE towards $NO_3RR$ remains at a high level. Meanwhile, the increased $NO_3RR$ activity as a result of the high nitrate concentration (as shown in Supplementary Fig. 30) can simultaneously facilitate the $NO_3RR$ over HER. As a result, the ammonia production in 1 M KOH and 0.5 M $KNO_3$ electrolyte is still the dominant reaction with a yield rate up to 26.6 mg $mg_{cat}^{-1}\,h^{-1}$ and an FE of 97.2% even at an applied potential of −0.4 V.

## Density functional theory (DFT) calculation

Based on previous findings that the reduction of $NO_3^-$ to NO is thermodynamically facile and that the activity and product selectivity and overall rate for nitrate reduction is determined by the major intermediate NO*[9,15,17,58,59], we focused on the free energy diagram of NO reduction to $NH_3$. Due to the great complexity of constructing the real catalyst surface under real experimental condition, the model was simplified here to the Co-doped MgO(111) and (Cu,Co)-doped

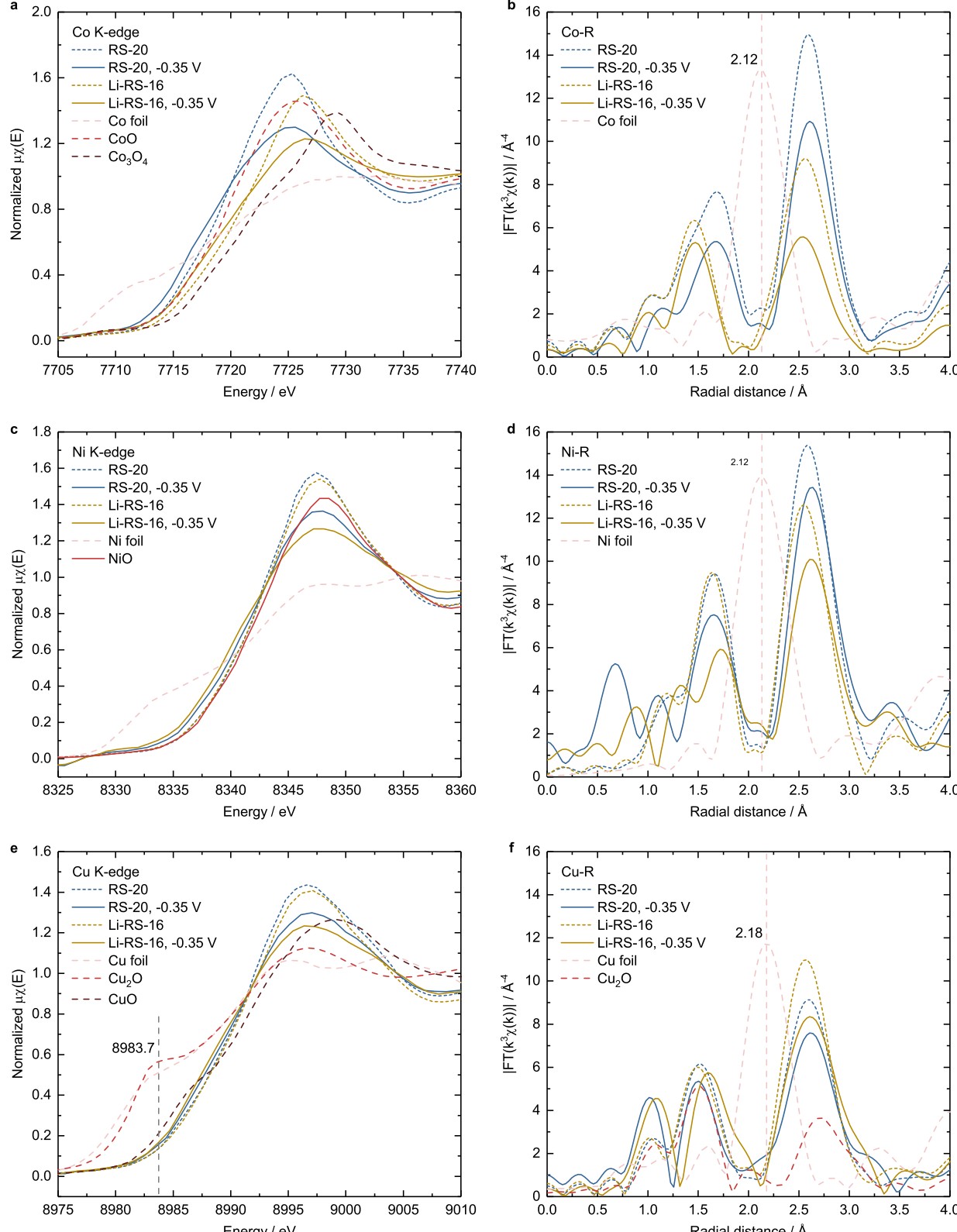

**Fig. 3 | The change of valence state and structure after nitrate reduction.** XANES of (**a**) Co, (**c**) Ni and (**e**) Cu K-edges and EXAFS of (**b**) Co, (**d**) Ni and (**f**) Cu of RS-20 and Li-RS-16 prior to and after nitrate reduction at −0.35 V for 30 min in 1 M KOH + 0.1 M KNO$_3$. Standard metals and metal oxides for reference data are attached in the figure, where the EXAFS intensities of Co, Ni and Cu foils are multiplied by 0.4.

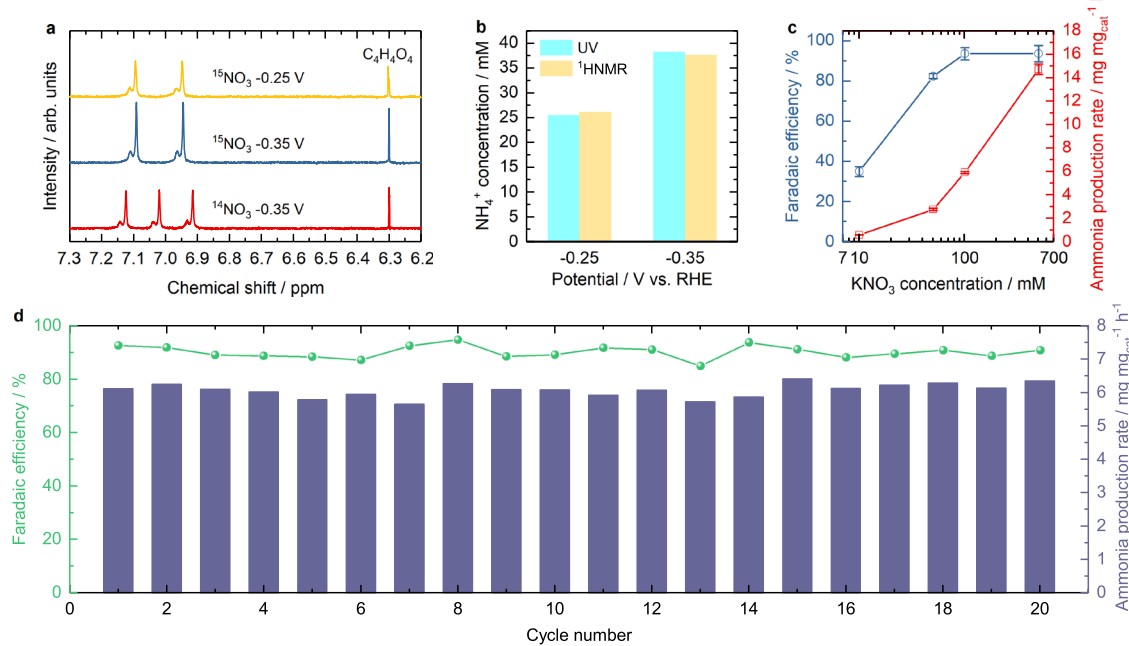

**Fig. 4 | Nitrate electroreduction on RS-20. a** $^1H$ NMR spectra of the electrolyte after $^{14}NO_3^-$ and $^{15}NO_3^-$ electroreduction. **b** Quantification of $^{15}NO_3^-$ electroreduction at −0.35 and −0.25 V by UV-vis and $^1H$ NMR spectra methods. **c** Nitrate concentration effect on ammonia production and FEs. **d** The cycling test. The standard deviations come from at least three independent measurements. Experimental condition: the electrode preparation and H-cell setup are the same with that in Fig. 1g, f. Each chamber had 18 mL of 1 M KOH and various concentrations of KNO₃ electrolyte saturated with Ar. The system resistance of the electrochemical cell containing 1 M KOH and various concentrations of KNO₃ is as follow. 0.01 M KNO₃, 0.979 ± 0.035 Ohm. 0.05 M KNO₃, 0.936 ± 0.034 Ohm. 0.1 M KNO₃, 0.828 ± 0.074 Ohm. 0.5 M KNO₃, 0.689 ± 0.045 Ohm.

MgO(111) surfaces (Supplementary Fig. 32), see the DFT Models rationale section below for details. Cu incorporation caused the increase of Co spin in both one-atom and two-atom doping modes (Figs. 6a, b). On the most stable Co-doped MgO(111) and (Cu,Co)-doped MgO(111) surfaces, the Co spin increased from 1.958 on the Co-doped MgO(111) to 2.158 on the (Cu,Co)-doped MgO(111). The increase in Co spin was found in all different Co/(Cu,Co)-doped MgO(111) structures studied (Supplementary Figs. 33, 34). NO adsorption was stronger on the (Cu,Co)-doped MgO(111) than on the Co-doped MgO(111) in all four possible NO adsorption structures (Supplementary Fig. 35). The free energy diagram for all possible pathways of NO reduction to $NH_3$ on the Co-doped MgO(111) and (Cu,Co)-doped MgO(111) surfaces were examined (Supplementary Figs. 36–38). The most stable NO adsorption occurred at the hexagonal closed packed hollow site, in which the N is bounded to two Co and one Mg on the Co-doped MgO(111) (surface structure in Fig. 6c), and with two Co and one Cu on the (Cu,Co)-doped MgO(111) (surface structure in Fig. 6d). The minimum energy pathway proceeds from NO* reduction to NOH*, further to N*, and step hydrogenation of N* to afford $NH_3$ on both surfaces. The potential limiting step is the reduction of $NH_2$* to $NH_3$* on both surfaces. At pH = 0, the free energy change ΔG ($NH_2$* → $NH_3$*) on the Co-doped MgO(111) surface was 0.87 eV, which is higher than 0.59 eV on the (Cu,Co)-doped MgO(111) surface. This result corresponds with the observation that Cu incorporation promotes ammonia generation. At pH = 14, the step from NH* to $NH_2$* becomes spontaneous, and ΔG ($NH_2$* → $NH_3$*) becomes 0.04 eV and −0.24 eV on the Co-doped MgO(111) and (Cu,Co)-doped MgO(111) surfaces, respectively. In addition, for the (Cu,Co)-doped MgO(111) surface, a high pH value also facilitated the transition from NO* to NOH*. These results show that Cu incorporation and a high pH value favor the transition from NO* → $NH_3$*.

Our results are also consistent with the recent theoretical findings by Chorkendorff, Nørskov and Wang[33–35] on spin promoted ammonia synthesis from $N_2$ in Haber-Bosch process. The nitrate reduction to

ammonia is a multi-step process, in which there is no N-N dissociation that occurs in Haber-Bosch process. The activity and selectivity trends of nitrate reduction on metals can be described by the adsorption strengths of atomic O and N. Due to the change of adsorbate from O to N and the intermediates during nitrate reduction, the adsorbate linear scaling relationships limit the potential maximum activity for single-site catalysts. In the Cu and Co combination, Cu favors reducing nitrate to nitrite and Co favors the selective reduction of N* to $NH_3$ due to strong nitrite binding[17]. The combination of Cu and Co breaks these scaling relations and reach the optimum point in the volcano plots and enhance the ammonia generation[9]. This enhancement from Cu and Co towards ammonia generation are reported in the alkaline[31,60] and neutral electrolyte[32,61]. However, in Haber-Bosch process, a gradually increasing adsorption energy makes the rate determining step of the reduction of $N_2$ to $NH_3$ shift from $N_2$ dissociation to $NH_x$ hydrogenation, and too strong nitrogen binding energy limits the hydrogenation of $NH_x$[35]. Recent works by Chorkendorff, Nørskov and Wang demonstrate the intermediates N adsorption depends on the spin state[33–35]. Take Co as an example, the spin-polarized Co has a weaker adsorption toward to N and N-N transition state than non-spin-polarized Co[33,34]. It means that the high spin of Co lifts the limit of the hydrogenation of $NH_x$ and enhances the ammonia generation. Thus, the recent theoretical findings on spin promoted ammonia synthesis in Haber-Bosch process also support our experimental observation that the higher spin state of Co facilitates the ammonia generation in the process of electrochemical nitrate reduction and our calculation results that higher spin state of Co reduces the barrier from $NH_2$ to $NH_3$. Our results can fall well into the magnetic enhancement effect and be fitted into the spin promoted ammonia synthesis picture.

In summary, we investigated the influence from spin states on Co-Cu pair on ammonia generation using rock-salt high-entropy oxides. Experimental results show a performance trend where RS-20 > Li-RS-16 > RS-0, which suggests a Cu-Co synergistic effect resulting in high ammonia generation. The surface reconstruction only occurs within a

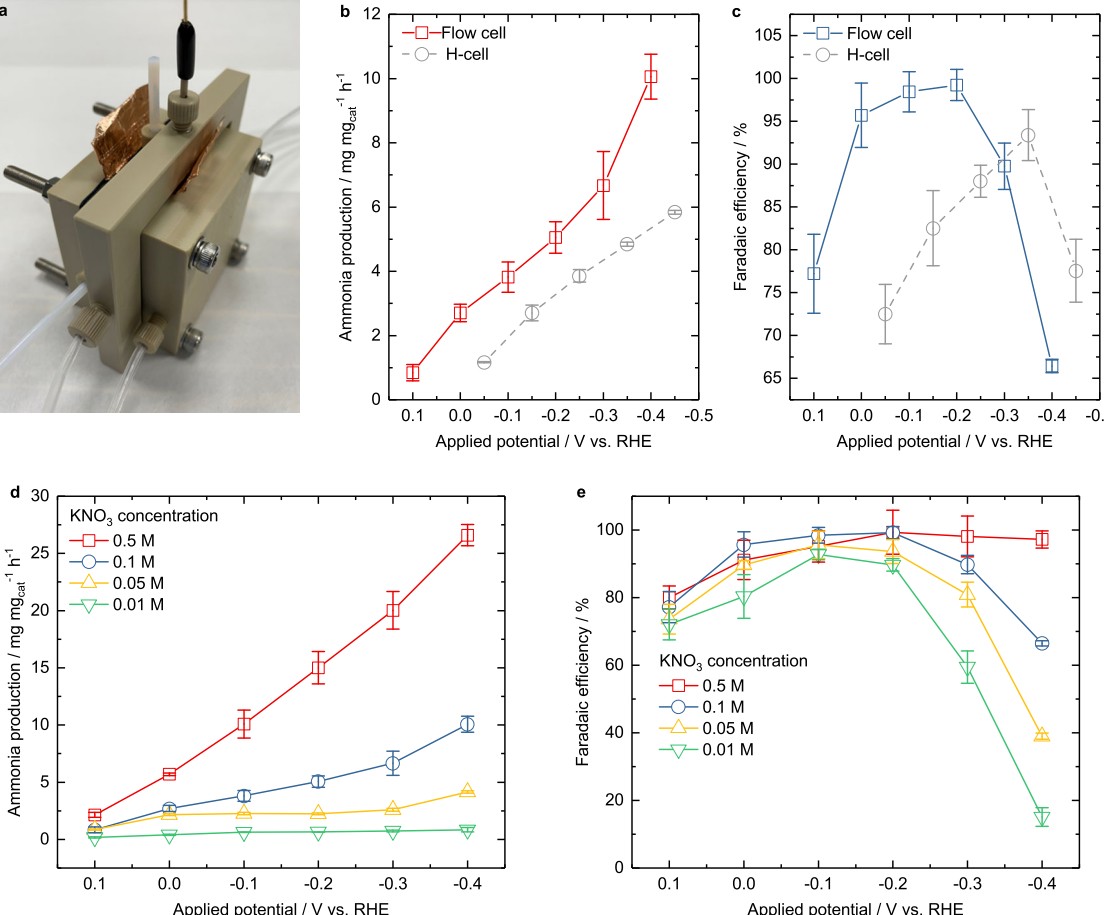

**Fig. 5 | Flow cell test. a** The set-up of the flow cell. The comparison of ammonia production (**b**) and FE (**c**) in the flow cell and H-cell with 1 M KOH and 0.1 M KNO$_3$ electrolyte. The ammonia production (**d**) and FE (**e**) with 1 M KOH and various nitrate concentrations in the flow cell. The standard deviations come from at least three independent measurements. Flow cell experimental condition: The mass loading of 0.5 mg$_{oxide}$ cm$^{-2}$ on the Sigracet 38BC (SGL Carbon) gas diffusion electrode, single side (2 × 2 cm$^2$); leak-free Ag/AgCl/3.4 M KCl electrode as reference electrode; each potential was hold for 30 min. The system resistance of the flow cell containing 1 M KOH and various concentrations of KNO$_3$ is as follow. 0.01 M KNO$_3$, 0.377 ± 0.021 Ohm. 0.05 M KNO$_3$, 0.343 ± 0.028 Ohm. 0.1 M KNO$_3$, 0.289 ± 0.012 Ohm. 0.5 M KNO$_3$, 0.295 ± 0.034 Ohm.

few nanometer-thick surfaces at some local area of RS-20, and hardly occurs on Li-RS-16 during nitrate reduction. Electronic state analyses reveal that the decrease in Co spin state weakens the Cu-Co synergistic effect. This result shows that the Co spin state should be considered when designing electrochemical processes for ammonia generation. Future studies will focus on the Co spin state contribution to ammonia generation by modulating the coordination environment in high-entropy oxides where cobalt cations are dispersed and in binary/ternary oxides where Co-O-Co coupling exists, and finally give a spin-related guide for high ammonia generation.

## Methods

### Material synthesis

The oxides Mg$_{0.25}$Co$_{0.25}$Ni$_{0.25}$Zn$_{0.25}$O (RS-0), Mg$_{0.20}$Co$_{0.20}$Ni$_{0.20}$Cu$_{0.20}$Zn$_{0.20}$O (RS-20), Li$_{0.20}$Mg$_{0.16}$Co$_{0.16}$Ni$_{0.16}$Cu$_{0.16}$Zn$_{0.16}$O (Li-RS-16) and Mg$_{0.40}$Ni$_{0.40}$Zn$_{0.20}$O (MNZO) were synthesized with the sol-gel method. All chemicals (lithium nitrate, magnesium nitrate hexahydrate, cobalt nitrate hexahydrate, nickel nitrate hexahydrate, copper nitrate hemi(pentahydrate), zinc nitrate hexahydrate, citric acid, and urea) were purchased from Sigma-Aldrich. The feeding mole ratio of metal, citric acid, and urea were 1:2:2. The metal precursors, citric acid, and urea were dissolved in de-ionized water with diluted nitric acid under stirring. The solution was kept at ~95 °C until the gel formed and then dried at 170 °C in the oven overnight, followed by

annealing at 1000 °C for 6 h at a 5 °C min$^{-1}$ ramping rate and cooling down naturally.

### Material characterization

X-ray diffraction (XRD, Bruker D8 Advance using Cu-Kα radiation) was used to characterize the crystal structure. Scanning electron microscope (SEM, JEOL JSM7600F) and transmission electron microscope (TEM, FEI Titan) were used to obtain the morphology and electron energy loss spectroscopy (EELS). X-ray photoelectron spectroscopy (XPS, Thermo Fisher Scientific Theta Probe) was used to assess the electronic state of these oxides, and the binding energy was calibrated from carbon contamination using C 1s peak at 284.8 eV. The extended X-ray absorption fine structure (EXAFS) and X-ray absorption near edge structure (XANES) were studied at the Singapore Synchrotron Light Source (SSLS), XAFCA beamline. Soft X-ray absorption experiment was also performed at the SUV beamline of SSLS. The vibrating-sample magnetometry (VSM) tests were carried at room temperature using Lake Shore 7400 VSM.

### Electrode preparation and electrochemical test

The ink was prepared by sonicating the oxide and ECP-600JD with mass ratio 4:1 in a mixed solution (volume ratio, deionized water: isopropanol: 5 wt% Nafion solution = 3:1:0.1344) until the ink became homogeneous. The ink was dropped on a glassy carbon electrode

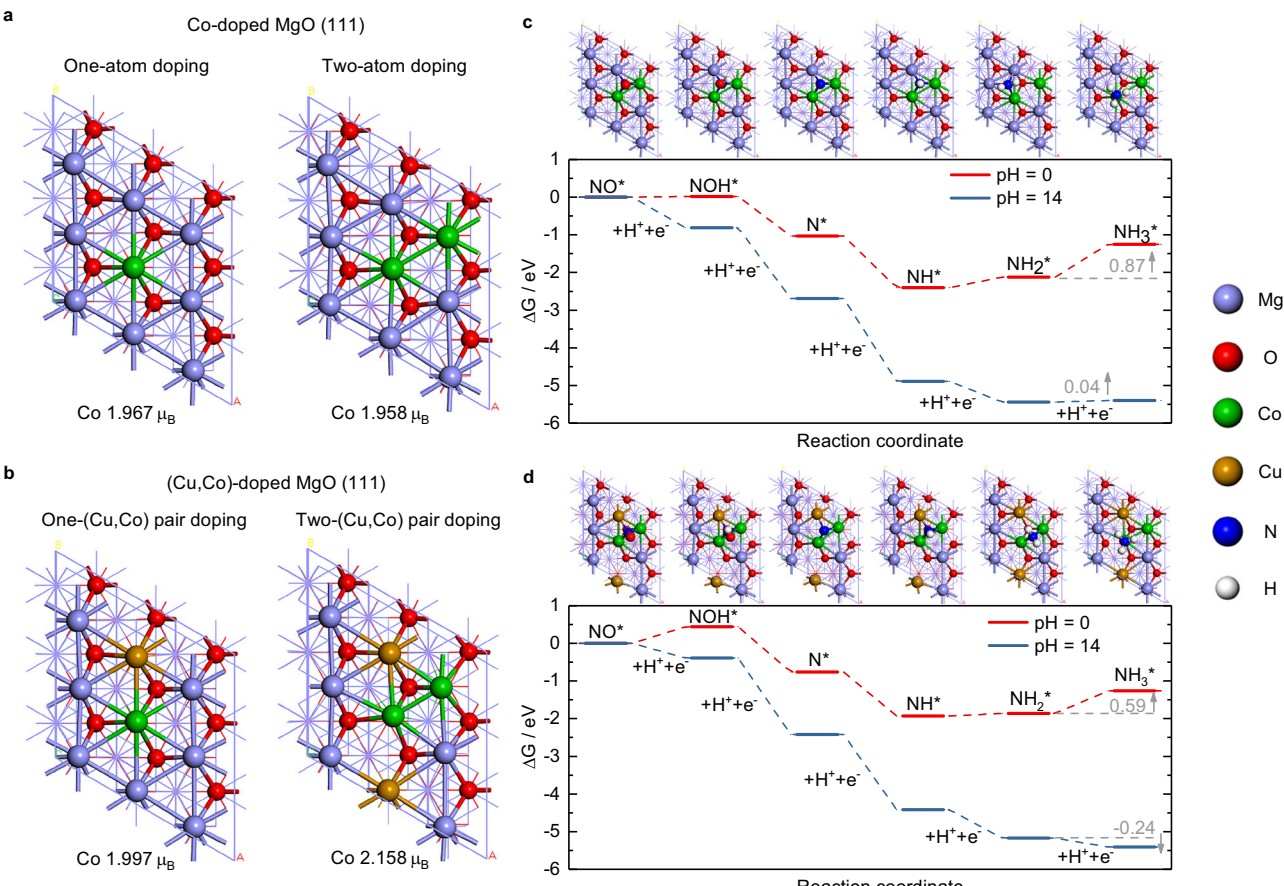

**Fig. 6 | DFT calculations.** The most stable (**a**) Co-doped MgO(111) and (**b**) (Cu,Co)-doped MgO(111) surface structures. Gibbs free energy in the minimum energy pathway of NO reduction to NH₃ at pH = 0 and pH = 14, U = 0.0 V/SHE on the (**c**) Co-doped MgO(111) surface and (**d**) (Cu,Co)-doped MgO(111) surface with the corresponding surface structures.

(diameter 5 mm) for linear sweep voltammetry (LSV) and cyclic voltammetry (CV) test. The mass loading is 1 mg$_{oxide}$ cm$^{-2}$. The reference and counter electrodes are Hg/HgO/1 M KOH electrode (0.098 V vs. standard hydrogen electrode, Tianjin Aida) and Pt plate (2 × 2 cm²). LSV tests were conducted in Ar-saturated 1 M KOH with or without 0.1 M KNO₃ at a scan rate of 10 mV s$^{-1}$ with 85% IR correction using Metrohm Autolab and Biologic SP-150 potentiostats. CV tests for the double-layer capacitance were conducted in 1 M KOH at the scan rates of 10, 20, 40, 60, 80, and 100 mV s$^{-1}$. All potential mentioned in the main text is versus the reversible hydrogen electrode (RHE). All reference electrodes were calibrated by measuring the open circuit potential (OCP) between the reference electrode and a standard hydrogen electrode (SHE) (Gaoss Union, China). The electrochemical impedance spectroscopy (EIS) was conducted in the frequency range from 100 kHz to 0.1 Hz at open circuit potential with an amplitude of 10 mV (root mean square (RMS)). The system resistance was estimated from the intersection with the real axis in the Nyquist plot. The standard deviations come from at least three independent measurements.

**Sample preparation for synchrotron test after nitrate reduction**
Considering the oxide adhesion to the carbon paper and oxide loading mass for synchrotron measurement, the ink was prepared by sonicating the oxide and ECP-600JD with mass ratio 4:1 in a mixed solution (volume ratio, deionized water: isopropanol: 5 wt% Nafion solution = 3:1:0.1344) until the ink became homogeneous. The ink was dipped onto both sides of Toray 090 carbon paper and each side dimensions are 7 × 7 mm². The total oxide mass loading is 30–40 mg cm$^{-2}$.

**Sample preparation for TEM-EELS after nitrate reduction**
For observing the change of the surface or bulk and decrease the disturbance from carbon in TEM observance, we used nickel foam to replace carbon as the current collector and conductive network. We imbedded less amount oxide powders into the nickel foam then press them for a closer contact between oxide powders and nickel foam. After nitrate reduction, sonicating the imbedded nickel form to obtain the oxide particles for TEM observation.

**Nitrate electrolysis and ammonia quantification by UV-vis spectra**
The Toray 090 carbon paper (1.5 × 1.5 cm²) supported catalyst with a mass loading of 1 mg$_{oxide}$ cm$^{-2}$ was used for nitrate electrolysis. The nitrate electrolysis tests were conducted at the designed potential for 30 min in a H-cell with FAA-3-PK-130 anion exchange membrane (FuMA-Tech) separating two chambers. The reference and counter electrodes are Hg/HgO/1 M KOH electrode and Pt plate. Each chamber had 18 mL of 1 M KOH and 0.1 M KNO₃ electrolyte saturated with Ar. The KNO₃ concentration 0.01, 0.05, 0.1 and 0.5 M were chosen for concentration response to ammonia production. The modified Indophenol blue method was used for ammonia quantification[15]. 2 mL of diluted product electrolyte was first mixed with 2 mL of 1 M NaOH solution containing 5 wt% salicylic acid and 5 wt% sodium citrate. 1 mL of 0.05 M NaClO solution and 0.2 mL of 1 wt% sodium nitroferricyanide solution were added subsequently. After 2 h, the absorption spectra of the resulting solution were acquired by UV-vis spectrophotometer (Agilent Technologies Cary 60 UV-Vis) and the absorbance at 657 nm was taken for calculation.

## Flow cell test

The electrochemical nitrate reduction reaction in a flow cell was conducted in a self-designed polyether ether ketone (PEEK) flow cell modified from the reference[13]. The flow cell consisted of the gas, catholyte and anolyte chambers. The gas and cathodic chambers were separated by a gas diffusion electrode (GDE) Sigracet 38BC (SGL Carbon). The catholyte and anolyte chambers were separated by an Fumasep FAA-3-PK-130 (Fumatech) anion-exchange membrane (AEM). The catalyst ink was prepared by sonicating the oxide and ECP-600JD with mass ratio 4:1 in a mixed solution (volume ratio, deionized water: isopropanol: 5 wt% Nafion solution = 3:1:0.1344) until the ink became homogeneous. The ink was air sprayed onto Sigracet 38BC (SGL Carbon) gas diffusion electrode on the single side. The area of the working electrode was $2 \times 2$ cm$^2$, and the catalyst loading was 0.5 mg$_{oxide}$ cm$^{-2}$. Catalysts loaded Sigracet 38BC (SGL Carbon) gas diffusion electrode, a leak-free Ag/AgCl/3.4 M KCl electrode (0.204 V vs. standard hydrogen electrode, Innovative Instruments) and a Pt plate ($2 \times 2$ cm$^2$) were employed as working, reference and counter electrodes, respectively. The thickness of the catholyte chamber was reduced to 4 mm only in our customized design to minimize the system resistance. The anolyte chamber dimensions are 2 cm × 2 cm × 1 cm. Prior to the electrolysis, Ar gas steam is used to purge the gas out of the electrolyte, 1 M KOH with various concentration (0.01, 0.05, 0.1 and 0.5 M) KNO$_3$ to avoid the possible oxygen reduction reaction. Then, 30 mL electrolyte was pumped into both the catholyte and anolyte chambers of the flow cell, respectively, with a flow rate of 5 mL min$^{-1}$ by (LongerPump, BT300-2J). The catholyte (30 mL) and anolyte (30 mL) were recycled in both chambers. Meanwhile, the Ar was purged into the gas chamber with a flow rate of 50 mL min$^{-1}$. The potential was applied at 0.1, 0, −0.1, −0.2, −0.3 and −0.4 V vs. RHE for 30 min, respectively for ammonia generation.

## Isotopic $^{15}NO_3^-$ labelling test

The nitrate electrolysis was conducted at −0.35 V and −0.25 V for 2 h. $^1$H NMR was used to determine the ammonia amount. Firstly, 5 mL of the product electrolyte was adjusted to pH = 2 by adding 0.5 M H$_2$SO$_4$. Then, 40 µL of 0.1 g mL$^{-1}$ maleic acid was added as an internal standard. 0.45 mL of the prepared solution and 0.05 mL of D$_2$O were mixed in the NMR tube for $^1$H NMR evaluation (JEOL JNM-ECA500II FT NMR System). pH measurements were performed using a Mettler Toledo 7easy pH Meter (U.S.). Prior to the measurement of the solutions, the pH meter was calibrated using Mettler-Toledo pH meter standard buffer solutions of pH 4.01, pH 7.00 and pH 9.21. Measurement of the pH was carried out on each freshly made solution prior to experiments.

## DFT method

Spin-polarized DFT + U calculations were conducted in this work, and the detailed method can be found in the Supplementary DFT method. The Gibbs free energy correction of gas molecules and adsorbed species was calculated using VASPKIT[62] at $T = 298.15$ K and $P = 1$ atm, except for gas H$_2$O molecule ($T = 298.15$ K and $P = 0.035$ atm). When the standard hydrogen electrode (SHE) is used as a reference electrode, the Gibbs free energy of (H$^+$ + e$^-$) pair at standard conditions (pH = 0, $T = 298.15$ K, and $P = 1$ atm) is equivalent to that of ½ H$_2$ in the gas phase[63]. At a pH different from 0, the Gibbs free energy of (H$^+$ + e$^-$) pair can be adjusted by the concentration dependence of the entropy, using the formula: $G(H^+ + e^-)_{pH} = G(H^+ + e^-)_{pH=0} + k_B T \ln(10)pH$[63,64].

## DFT models

Because the role of Mg, Ni, and Zn in stabilizing the structures of HEOs was deduced from experimental results, MgO was used to represent (Mg,Ni,Zn)O HEO in this theoretical investigation. Therefore, here, the Co-doped MgO(111) and (Cu,Co)-doped MgO(111) were used to represent the Mg$_{0.25}$Co$_{0.25}$Ni$_{0.25}$Zn$_{0.25}$O and

Mg$_{0.20}$Co$_{0.20}$Ni$_{0.20}$Cu$_{0.20}$Zn$_{0.20}$O surface, respectively. The lattice constant of MgO was calculated to be 4.234 Å, which is close to the experimental value (4.21 Å)[65]. A p(3×3) MgO(111) surface model with ten atomic layers was constructed first, where the top six layers including adsorbates were allowed to relax, and the bottom four layers were fixed in their bulk position, as shown in Supplementary Fig. 32. The vacuum region between periodically repeated slabs was set at 15 Å. In the Co-doped MgO(111), two Mg atoms of the first and/or second layer were replaced with two Co atoms. In the (Cu,Co)-doped MgO(111), two Mg atoms in the first and/or second layer around the Co dopants were replaced by two Cu atoms.

## DFT models rationale

Experimental results show that the Mg, Ni, and Zn in HEOs are not active sites but rather play a role in stabilizing HEO structures. In DFT simulations, the Hubbard + U correction (DFT + U) need to be employed to describe strong electron correlation effects in NiO and ZnO, but it is not required in MgO. Therefore, to simplify DFT simulations of (Mg,Ni,Zn)O HEO, we used MgO, taking into account computational efficiency and methodological accuracy. In addition, there are two reasons for the choice of MgO(111): (1) in our experiments, pH is close to 14. It was reported that the hydroxylated (111) surface has a lower surface energy than MgO(100) and MgO(110)[66]. (2) The MgO(111) surface model (with a specific termination) we used in this work has been well studied in the literature[67]. On the MgO(111) surface, all possible substitutional sites of Cu and Co dopants were calculated (Supplementary Figs. 33, 34), including the surface and subsurface, and the most stable structures were then chosen to represent the experimental electrocatalysts.

## Data availability

Additional data are available from the corresponding author upon request.

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

## Acknowledgements

This work was supported by the Agency for Science, Technology and Research (Central Research Fund Award, Z.W.S), and the A*STAR AME IAF-PP (Grant No. A19E9a0103, J.Z.). It was partially supported by the Agency for Science, Technology and Research (A*STAR) MTC Individual Research Grants (IRG) M22K2c0078, Z.J.X and the Singapore National Research Foundation under its Campus for Research Excellence and Technological Enterprise (CREATE) programme. We also acknowledge high-performance computational facilities from the National Supercomputing Centre (NSCC) Singapore (https://www.nscc.sg) and A*STAR Computational Resource Centre (A*CRC).

## Author contributions

S.S., C.D., P.Z. contributed equally to this work. S.S., Z.J.X., Z.W.S. conceived the original concept and initiated the project. S.S. wrote the manuscript. C.D., P.Z., J.Z., Z.J.X., Z.W.S. revised it. S.S. synthesized the materials and performed the characterization with assistance from S.X. (XANES and EXAFS), Y.R. (XPS), H.R.T. (TEM), P.C.L. (XRD), M.L. (TEM), C.D. (soft X-ray absorption), D.Z. (conductivity measurement), C.W. (XANES and EXAFS), A.Y. (VSM), W.Y.L. (XPS) and D.H.L.S. (XPS). C.D., L.S. designed the flow cell. C.D., J.C.J.K. conducted the electrochemical test and analyzed the electrochemical, UV-vis, and NMR data. P.Z. carried out the theoretical calculation and results analysis. Y.L., T.L.T. contributed to the analysis and discussion of simulation results. J.Z. supervised the simulation work.

## Competing interests

The authors declare no competing interests.
