## [Peer Review File · Nature Communications]

REVIEWER COMMENTS

Reviewer #1 (Remarks to the Author):

The authors have proposed a high-entropy oxide ($\text{Mg}_{0.2}\text{Co}_{0.2}\text{Ni}_{0.2}\text{Cu}_{0.2}\text{Zn}_{0.2}\text{O}$, RS-20) to accelerate electrochemical nitrate-to-ammonia conversion. The RS-20 exhibits good catalytic performance (Faradaic efficiency 99.3%, yield rate 26.6 mg $\text{mgcat}^{-1}\text{h}^{-1}$). The author explained that the coexistence of Cu and Co is crucial for achieving a high spin state in Co needed for improved ammonia generation. However, some concerns should be taken into consideration to further improve the level of this work.

1. The author claimed that 'the catalytic performance achieved in this work (Faradaic efficiency 99.3%, yield rate 26.6 mg $\text{mgcat}^{-1}\text{h}^{-1}$) is among the best in the literature to date'. However, the better catalytic performance has been reported (Nature Nanotechnology, 2022, 17, 759-767).
2. The spin state was confirmed by XPS and Spin-polarized DFT calculations. More experimental evidence of the spin state should be explored. The VSM (Vibrating Sample Magnetometer) results will be helpful.
3. Which is the actual active site? Co, Cu, or Co and Cu?
4. What is the role of Mg, Zn and Mg species in the RS-20?
5. Why the DFT model can be simplified to MgO (111) surface with Co and Cu-doping?

Reviewer #2 (Remarks to the Author):

This work offers innovative perspectives for studying ammonia production through nitrate reduction by considering the spin state on high entropy oxide MgCoNiCuZnO . The idea of isolating homo-cations to impede the coupling of N atoms to N_2 and H atoms to H_2 is constructive.

The authors highlighted the synergistic effect of the Co-Cu pair within the high-entropy oxide and how the introduction of Li causes the deactivation of this synergistic effect. This finding is of significant importance as it challenges established opinions in the field. This study offers a unique and valuable contribution to the understanding of the effects of spin state on nitrate reduction processes. In addition, the authors made an important observation regarding the electrochemical reconstruction on the high entropy oxide during nitrate reduction. They found that this reconstruction process is minimal or nearly absent, which contrasts with previous reports on Cu/Co compounds where surface/bulk evolution plays a significant role in promoting nitrate reduction. This distinction is noteworthy and adds further value to the research findings.

Thus, I believe that this manuscript is highly recommendand for being published in Nat. Commun. after addressing the following minor concerns.

1. Regarding the nitrate reduction activity comparison, it would be beneficial for the authors to include the nitrate reduction activity of $\text{Mg}_{0.25}\text{Ni}_{0.25}\text{Cu}_{0.25}\text{Zn}_{0.25}\text{O}$ with the rocksalt structure. This additional data will allow for a more comprehensive comparison between different compositions with the same structures.

2. It is intriguing to explore the activity of Li-doped high entropy oxides at varying concentrations. Requesting information on the activity of a few Li-doped cases, in addition to the comparison between $\text{Mg}_{0.25}\text{Co}_{0.25}\text{Ni}_{0.25}\text{Zn}_{0.25}\text{O}$ and $\text{Li}_{0.20}\text{Mg}_{0.16}\text{Co}_{0.16}\text{Ni}_{0.16}\text{Cu}_{0.16}\text{Zn}_{0.16}\text{O}$, could provide insights into the role of Li and its concentration-dependent effects.

3. Although the authors have presented edx line scans demonstrating minimal changes on the surface during electrolysis, it would be valuable to provide surface information prior to the electrolysis process. This data would contribute to a better understanding of the initial surface state and facilitate a more comprehensive analysis of the electrochemical processes occurring during nitrate reduction.

4. Expanding the discussion on the relationship between spin state and nitrate reduction would enhance the understanding of the proposed mechanism. Elaborate on how the spin state affects the catalytic activity and elaborate on any previous studies or theoretical frameworks that support the role of spin state in nitrate reduction processes.

5. Drawing a comparison between high entropy oxides and single atoms in terms of the feature of isolated homo-cations would be insightful. Discuss the similarities and differences between these two systems, highlighting their respective advantages and potential applications in catalysis.

6. Comparing and discussing the stability of high entropy oxides and monoxides in terms of surface construction would provide valuable insights. Elaborate on the stability of high entropy oxides under the conditions of nitrate reduction and compare it to the stability of monoxides.

Reviewer #3 (Remarks to the Author):

The authors test two high entropy oxides, MgCoNiCuZnO and LiMgCoNiCuZn for nitrate reduction to ammonia. They also test an oxide without Cu which also has low activity. The oxide without Li has a higher activity, which the authors attribute to high spin Co compared to low spin Co, where the low spin decreases the activity. This is based on electronic structure measurements. The concept of controlling beyond the elemental composition is interesting, but it is not clear to me that this is definitively shown to be a reason for enhancement on these systems. In addition, the per site activity of the catalysts is unclear in relation to other catalysts reported in the literature. Therefore, although the concept of new ways to control and go beyond scaling relation limitations of adsorption energies is interesting, it is not clear whether this publication clearly shows this. In addition, there are some details that need to be included to understand the results, particularly the experimental conditions, which are known to significantly impact the performance for nitrate reduction.

An additional positive is the use of the flow cell to control mass transport, as there are only a few studies that have operated NO₃RR in a flow cell (ref 13 cited being one of them I believe). The use of this to control mass transport and operate in continuous mode is interesting, but as mentioned below brings into question some of the fundamental insights. Regardless, I did find this aspect to be novel in the NO₃RR literature and perhaps something that should be done more often. As it is a somewhat interesting component, I found the description of the cell and its operating conditions a bit confusing and it could be elaborated on.

The authors probe the spin state in part based on spin-related work from Chorkendorff and Norskov recently for promoters of Haber Bosch process (magnetic enhancement and electrostatic). It is not completely clear to me from the manuscript whether the authors believe the effects of the spin state fall into the magnetic enhancement effects or electrostatic effects. A more detailed discussion of the recent paper by Chorkendorff and Norskov, and how the results here fit into this picture is needed. As it is written, it is hard to interpret without doing significant reading of other literature, some of which is quite recent.

On a per site basis how good are the catalysts compared to similar conditions (cations, anions in electrolyte, pH, potential, temperature all kept constant, nitrate concentration constant)? For example, in the sentence "Our ammonia generation performance is comparable to other works (Table S1), despite the differences in electrode preparation, applied potential, solution pH value, nitrate concentration, and normalizations of performance indicators." I do not understand why this would be relevant, since it may just be a coincidence that the activity is the same, as all of those parameters are well-known to affect the ammonia generation.

In my opinion, it is very difficult to deconvolute the effects that contribute to the nitrate reduction activity and selectivity. In particular, having two samples makes it a challenge as there are multiple factors being varied. A stronger case of the spin state would be to have control of it over a series of values and show how the activity correlates, rather than only three data points (RS-0, RS-20 and Li-RS-16). The authors compare and imply that RS-0 and RS-20 are evidence of the spin state being a factor (RS-0 has no Cu, and RS-20 has higher spin state), but it is unclear how the Cu presence may affect this.

Is there a way to tune the amount of Li so that the spin state of the Co can be systematically varied and plotted against the activity under controlled conditions?

Can the authors compare to Cu and Co (and CuCo alloy) controls under the same conditions to show whether there is an enhancement? Ideally with controlled mass transport to avoid any artifacts and with surface area or site normalization. It is hard to understand what is meant by “high-entropy oxide RS-20 is a decent candidate”.

For the characterization (XANES, etc), how relevant is it to the structure under reaction conditions (i.e. after reconstruction) and how relevant is the bulk structure from XANES/EXAFS to the surface structure that is important for electrocatalysis?

For the reduction of the oxides, could the electrical conductivity of the oxides be reported? Is it an issue of electrical conductivity why the bulk cannot be reduced? I am unfamiliar with these oxides and their use as electrocatalysts. How much does the pure conductivity change for the different oxides and how would this influence the results?

I cannot comment in detail on the DFT, apart from saying that it is uncertain how the structures used for the calculations match the actual structure under operating condition of the experimental electrocatalyst.

General reporting

Reference voltage should be mentioned in all places in the text, as well as electrolyte. The electrolyte plays a huge role on performance and it is not made clear what conditions these are studied in. Is there a change in the pH during operation? If so it should be clearly stated.

What is the reason for the decrease in FE at high overpotentials in Figure 1h? It is not sufficiently explained or discussed.

There is insufficient detail in the figure captions. For example, Figure 1 does not include the nitrate concentration, an enormously large factor in nitrate reduction current density.

In the discussion and results when the flow cell was used, this seems to imply earlier results may have been transport limited? This means that the trends observed may not be valid in the H-cell, because they were transport limited. This needs to be addressed.

For the flow cell, could the current for applied case vs. geometric current density also be reported? This would be helpful for context and reference. Also, in the methods I did not quite understand what the gas was for, why is a gas stream needed? I assumed that pure nitrate liquid electrolyte would be fed to the cathode.

Point-by-point response to Reviewer #1's comments

The authors have proposed a high-entropy oxide ($\text{Mg}_{0.2}\text{Co}_{0.2}\text{Ni}_{0.2}\text{Cu}_{0.2}\text{Zn}_{0.2}\text{O}$, RS-20) to accelerate electrochemical nitrate-to-ammonia conversion. The RS-20 exhibits good catalytic performance (Faradaic efficiency 99.3%, yield rate $26.6 \text{ mg mg}_{\text{cat}}^{-1} \text{ h}^{-1}$). The author explained that the coexistence of Cu and Co is crucial for achieving a high spin state in Co needed for improved ammonia generation. However, some concerns should be taken into consideration to further improve the level of this work.

1. The author claimed that ‘the catalytic performance achieved in this work (Faradaic efficiency 99.3%, yield rate $26.6 \text{ mg mg}_{\text{cat}}^{-1} \text{ h}^{-1}$) is among the best in the literature to date’. However, the better catalytic performance has been reported (*Nature Nanotechnology*, 2022, 17, 759-767).

Response 1: We thank Reviewer #1 for pointing out this issue. We removed the statement “the catalytic performance achieved in this work (Faradaic efficiency 99.3%, yield rate $26.6 \text{ mg mg}_{\text{cat}}^{-1} \text{ h}^{-1}$) is among the best in the literature to date” in the revised manuscript.

We have cited this work (*Nat. Nanotechnol.*, 2022, 17, 759-767) in the Introduction part, which has been highlighted in the revised manuscript as below.

“Alloying a second metal such as Zn^{19} , Sn^{20} , or Ni^{13} is a common approach to modulate the Cu d-band center and the intermediate adsorption energy to increase ammonia generation. Additionally, due to Cu having the inert hydrogen evolution activity, Cu can also be used as a matrix for dispersing Ru with low nitrate activation barrier for highly ammonia selectivity and production²¹.”

We also list this work for the comparison with other Cu and Co-based compounds in the revised Supplementary Table 4 as below.

	Catalyst	electrolyte	NH ₃ FE	NH ₃ yield rate	potential	Year / ref
1	Mg _{0.20} Co _{0.20} Ni _{0.20} Cu _{0.20} Zn _{0.20} O	1 M KOH + 0.1 M NO ₃ ⁻ 1 M KOH + 0.5 M NO ₃ ⁻	99.3% 97.2%	5.05 mg mg _{cat} ⁻¹ h ⁻¹ 26.6 mg mg _{cat} ⁻¹ h ⁻¹	-0.2 V vs. RHE -0.4 V vs. RHE	This work
2	Ru-Cu nanowire	1 M KOH + 0.032 M NO ₃	96.0%	76.563 mg h ⁻¹ cm ⁻²	0.04 V vs. RHE	2022 ³⁰
3	Cu ₅₀ Co ₅₀ / nickel foam	1 M KOH + 0.1 M KNO ₃	~100%	4.8 mmol cm ⁻² h ⁻¹	-0.2 V vs. RHE	2022 ³¹
4	Co _{0.5} Cu _{0.5} / carbon fiber	1 M KOH + 0.05 M NO ₃ ⁻	>95%	176 mA cm ⁻²	-0.03 V vs. RHE	2022 ³²
5	Ar-plasma treated Cu ₃₀ Co ₇₀ / carbon paper	1 M KOH + 0.1 M NO ₃ ⁻	~80%	5.13 mg cm ⁻² h ⁻¹	-0.47 V vs. RHE	2022 ³³
6	Cu-Co binary sulfides evolved Cu/CuO _x -Co/CoO hybrids	0.1 M KOH + 0.01 M NO ₃ ⁻	93.3%	1.17 mmol cm ⁻² h ⁻¹	-0.175 V vs. RHE	2022 ²⁸
7	ZnCo ₂ O ₄ / carbon paper	0.1 M KOH + 0.1 M NO ₃ ⁻	95.4%	2.10 mg mg ⁻¹ h ⁻¹	-0.4 V vs. RHE	2022 ³⁴
8	CoO/N-doped carbon nanotube/graphite paper	0.1 M KOH + 0.1 M NO ₃ ⁻	93.8%	9.04 mg h ⁻¹ cm ⁻²	-0.6 V vs. RHE	2022 ³⁵
9	Ultrathin CoO _x nanosheets	0.1 M KOH + 0.1 M NO ₃ ⁻	93.4%	82.4 mg h ⁻¹ mg ⁻¹	-0.3 V vs. RHE	2021 ²⁹
10	Cu ₅₀ Ni ₅₀ / PTFE	1 M KOH + 0.1 M NO ₃ ⁻	99%	-	-0.15 V vs. RHE	2020 ³⁶
11	Cu single atom	0.1 M KOH + 0.1 M NO ₃ ⁻	84.7%	4.5 mg cm ⁻² h ⁻¹ / 12.5 mol g _{Cu} ⁻¹ h ⁻¹	-1.0 V vs. RHE	2022 ³⁷
12	Cu(100)-rich rugged Cu-nanobelt	1 M KOH + 0.1 M NO ₃	95.3%	650 mmol h ⁻¹ g _{cat} ⁻¹ h ⁻¹	-0.15 V vs. RHE	2021 ²⁷
13	Cu ₂₊₁ O/Ag- carbon cloth	0.1 M KOH + 0.01 M KNO ₃	85.03%	2.2 mg h ⁻¹ cm ⁻²	-0.74 V vs RHE	2023 ³⁸
14	Cu-doped Fe ₃ O ₄	0.1 M KOH + 0.1 M KNO ₃	~100%	179.55 mg h ⁻¹ mg _{cat} ⁻¹	-0.6 V vs RHE	2023 ³⁹
15	Cu-modified Ru/C	0.1 M NaOH + 0.1 M NaNO ₃	95%	23.7 μmol h ⁻¹ cm ⁻² 6.86 mmol h ⁻¹ mg _{Ru} ⁻¹	-0.1 V vs. RHE	2023 ⁴⁰
16	Cu@nickel foam	1 M KOH + 200 ppm nitrate-N	96.6%	0.252 mmol h ⁻¹ cm ⁻²	-0.23 V vs. RHE	2021 ⁴¹

2. The spin state was confirmed by XPS and Spin-polarized DFT calculations. More experimental evidence of the spin state should be explored. The VSM (Vibrating Sample Magnetometer) results will be helpful.

Response 2: We thank Reviewer #1 for this advice. We conducted the VSM test (Lake Shore 7400 VSM) and soft X-ray absorption for $\text{Mg}_{0.25}\text{Co}_{0.25}\text{Ni}_{0.25}\text{Zn}_{0.25}\text{O}$ (RS-0), $\text{Mg}_{0.20}\text{Co}_{0.20}\text{Ni}_{0.20}\text{Cu}_{0.20}\text{Zn}_{0.20}\text{O}$ (RS-20) and $\text{Li}_{0.20}\text{Mg}_{0.16}\text{Co}_{0.16}\text{Ni}_{0.16}\text{Cu}_{0.16}\text{Zn}_{0.16}\text{O}$ (Li-RS-16) for providing more experimental evidence of the spin state. The relevant discussion can be found in the description following the revised Supplementary Figs. 14-15.

Figure R#1-1 shows the M-H curves of RS-0, RS-20 and Li-RS-16 at room temperature, where M is the magnetization, H is the magnetic field strength. The magnetic susceptibility χ can be obtained based on the equation $M = \chi H$. The linear M-H relationship and the positive χ values of RS-0, RS-20, Li-RS-16 indicate that these three samples are paramagnetic at room temperature. From the relationship $\chi = N g^2 J(J+1) \mu_B^2 / (3k_B T)$, where N, g, J, μ_B , k_B and T are the Avogadro number, g-factor, angular momentum quantum number, Bohr magneton, Boltzmann constant and temperature, respectively. A larger χ value means a larger J, indicating a higher spin state. The M-H curves shows that RS-0 and RS-20 have a very close magnetic susceptibility. Li-RS-16 has a smaller magnetic moment than RS-0 and RS-20. The total mole fractions of Co, Ni and Cu in RS-0, RS-20 and Li-RS-16 are 0.50, 0.60 and 0.48, respectively. If the magnetic moment contribution is merely from Co, Ni and Cu, the magnetic moments of RS-0 and RS-20 are still higher than that of Li-RS-16. Because the electronic states of Ni and Cu are not observed to be remarkably different, the magnetic moment decreases in Li-RS-16 could come from the decrease in Co spin state.

Fig. R#1-1. The M-H curves of RS-0, RS-20 and Li-RS-16 at room temperature, (a) magnetic moment normalized by (a) mass and (b) total mole numbers of Co, Ni and Cu. We have added it into the revised Supplementary Fig. 14.

Apart from VSM, we also used the soft X-ray to characterize the Co, Ni and Cu L-edges and O K-edge of RS-0, RS-20 and Li-RS-16 (Fig. R#1-2). Soft X-ray absorption experiments were performed at the SUV beamline of Singapore Synchrotron Light Source (SSLS). The $L_{2,3}$ -edges of Co, Ni and Cu represent the transition from $2p^63d^n$ to $2p^53d^{n+1}$, where n is the electron number in d orbitals. Take Co as an example, lower energy L_3 peak stands for the transition $2p_{3/2} \rightarrow 3d$ and higher energy L_2 peak stands for the transition $2p_{1/2} \rightarrow 3d$ (*Hibberd et al., J. Phys. Chem. C 2015, 119, 4173–4179*). The Co L_3 peaks of RS-0 and RS-20 splits into two peaks at 780.8 eV and 782.0 eV with similar weight. The Co L_3 peak of Li-RS-16 splits into a weak peak at 780.8 eV and a strong peak 782.5 eV. Although the interpretation of Co L-edge requires the consideration of multiplet splitting, hybridization, and crystal field effects, here we simplify this behavior as energy splitting between the t_{2g} states and the e_g states (*Kroll et al., Phys. Rev. B, 2006, 74, 115123*). In the O_h symmetry, Co^{2+} has the low spin state $t_{2g}^6e_g^1$ and high spin state $t_{2g}^5e_g^2$, and Co^{3+} has the low spin state $t_{2g}^6e_g^0$, intermediate spin state $t_{2g}^5e_g^1$, and high spin state $t_{2g}^4e_g^2$. Therefore, the peak intensity theoretically corresponds to the hole number of t_{2g} (at low energy) and e_g (at high energy). The Co L_3 -edges of RS-0 and RS-20 have no obvious change, indicating no electron occupation change. When comparing Li-RS-16 and RS-20, the peak corresponding to e_g orbitals shifts to the high energy direction relative to RS-20. This indicates that the Co valence state increases, in agreement with our XANES results and the previous investigation (*Chin et al., Phys. Rev. B, 2019, 100, 205139*). In Li-RS-16, the peak corresponding to e_g is much stronger than the peak corresponding to t_{2g} , it means that less holes are in t_{2g} orbitals and more holes are in e_g orbitals. For Co^{2+} or Co^{3+} , it suggests there are less unpaired electrons and exhibits lower spin. Besides, there is a larger peak splitting energy between t_{2g} and e_g orbitals in Li-RS-16 than that in RS-0 and RS-20, which is consistent with EXAFS results that Co-O distance is smaller in Li-RS-16 than that in RS-0 and RS-20. A smaller Co-O distance causes a strong crystal field that induces a larger t_{2g} and e_g splitting.

For Ni and Cu cations, we did not observe the obvious difference among RS-0, RS-20 and Li-RS-16. Since the O K-edge represents the electronic transitions from the O 1s core level to the unoccupied transition metal 3d levels with the O 2p components, due to the coexistence of multiple cations, we cannot use it to analyse the d orbital occupation of cations. A remarkable difference is that the signal reflecting the hole in oxygen (at ~531.5 eV) becomes stronger from RS-0, RS-20 to Li-RS-16, which is consistent with our EELS results. Another difference is the disappearance of the peak at 533.2 eV representing the decrease of metal-oxygen covalency (*Suntivich et al., J. Phys. Chem. C 2014, 118, 1856-1863*), which could be due to the highly ionic Li-O bond.

Fig. R#1-2. The (a) Co L-edge, (b) Ni L-edge, (c) Cu L-edge and (d) O K-edge XAS of RS-0, RS-20 and Li-RS-16. We have added it into the revised Supplementary Fig. 15.

3. Which is the actual active site? Co, Cu, or Co and Cu?

Response 3: We thank Reviewer #1 for raising this question. Both Co and Cu are active sites. From those reports on nitrite reduction on individual Co-based oxides (*Wang et al., ACS Catal. 2021, 11, 15135–15140*) and Cu-based oxides (*Daiyan et al., Energy Environ. Sci., 2021, 14, 3588–3598*), both Co and Cu can reduce nitrate. The coexistence of Co and Cu can enhance the nitrate reduction activity based on the favourable reduction from nitrate to nitrite on Cu and from nitrite to ammonia on Co (*Fu et al., J. Hazard. Mater., 2022, 434, 128887*). From our experiments results, $\text{Mg}_{0.25}\text{Co}_{0.25}\text{Ni}_{0.25}\text{Zn}_{0.25}\text{O}$, CuO , and $\text{Mg}_{0.20}\text{Co}_{0.20}\text{Ni}_{0.20}\text{Cu}_{0.20}\text{Zn}_{0.20}\text{O}$ can reduce the nitrate in the revised Supplementary Figs. 7, 10. When Cu and Co coexist in the catalysts, a better nitrate reduction activity can be observed.

4. What is the role of Mg, Zn and Mg species in the RS-20?

Response 4: We thank Reviewer #1 for raising this question. The role of Mg is to stabilize the single rock-salt phase, which has been reported in the work “High entropy oxides for reversible energy storage” (*Sarkar et al., Nature Communications, 2018, 9, 3400*), in which Mg-free $\text{Co}_{0.25}\text{Cu}_{0.25}\text{Ni}_{0.25}\text{Zn}_{0.25}\text{O}$ cannot stabilize the single rock-salt phase. We attempted to synthesize the Zn-free $\text{Mg}_{0.25}\text{Co}_{0.25}\text{Ni}_{0.25}\text{Cu}_{0.25}\text{O}$ by the same method reported in our work and found the composition $\text{Mg}_{0.25}\text{Co}_{0.25}\text{Ni}_{0.25}\text{Cu}_{0.25}\text{O}$ has a secondary CuO phase and cannot stabilize the single rock-salt phase (Fig. R#1-3). In our work, we need a single rock-salt phase that can stabilize both Co and Cu simultaneously. Thus, the role of Mg and Zn was to stabilize the single rock-salt phase.

Fig. R#1-3. The XRD pattern of $\text{Mg}_{0.25}\text{Co}_{0.25}\text{Ni}_{0.25}\text{Cu}_{0.25}\text{O}$.

5. Why the DFT model can be simplified to MgO (111) surface with Co and Cu-doping?

Response 5: We would like to thank you for pointing out this. Detecting and understanding of catalyst structure in operando conditions at the atomic level remains a big problem in electrocatalysis. In this work, the experimental electrocatalyst was simplified to MgO(111) with Co and Cu-doping in the DFT model, considering the computational efficiency and methodological accuracy. We added additional description to about this issue in the revised manuscript (highlighted) as below.

“DFT Models rationale. Experimental results show that the Mg, Ni, and Zn in HEOs are not active sites but rather play a role in stabilizing HEO structures. In DFT simulations, the Hubbard + U correction (DFT + U) need to be employed to describe strong electron correlation effects in NiO and ZnO, but it is not required in MgO. Therefore, to simplify DFT simulations of (Mg,Ni,Zn)O HEO, we used MgO, taking into account computational efficiency and methodological accuracy. In addition, there are two reasons for the choice of MgO(111): (1) in our experiments, pH is close to 14. It was reported that the hydroxylated (111) surface has a lower surface energy than MgO(100) and MgO(110) (*Geysermans et al., Phys. Chem. Chem. Phys., 2009, 11, 2228–2233*). (2) The MgO(111) surface model (with a specific termination) we used in this work has been well studied in the literature (*Zhang and Tang, J. Phys. Chem. C 2008, 112, 3327-3333*). On the MgO(111) surface, all possible substitutional sites of Cu and Co dopants were calculated (Supplementary Figs. 33-34), including the surface and subsurface, and the most stable structures were then chosen to represent the experimental electrocatalysts.”

Point-by-point response to Reviewer #2's comments

This work offers innovative perspectives for studying ammonia production through nitrate reduction by considering the spin state on high entropy oxide MgCoNiCuZnO. The idea of isolating homo-cations to impede the coupling of N atoms to N₂ and H atoms to H₂ is constructive.

The authors highlighted the synergistic effect of the Co-Cu pair within the high-entropy oxide and how the introduction of Li causes the deactivation of this synergistic effect. This finding is of significant importance as it challenges established opinions in the field. This study offers a unique and valuable contribution to the understanding of the effects of spin state on nitrate reduction processes. In addition, the authors made an important observation regarding the electrochemical reconstruction on the high entropy oxide during nitrate reduction. They found that this reconstruction process is minimal or nearly absent, which contrasts with previous reports on Cu/Co compounds where surface/bulk evolution plays a significant role in promoting nitrate reduction. This distinction is noteworthy and adds further value to the research findings.

Thus, I believe that this manuscript is highly recommendand for being published in Nat. Commun. after addressing the following minor concerns.

1. Regarding the nitrate reduction activity comparison, it would be beneficial for the authors to include the nitrate reduction activity of Mg_{0.25}Ni_{0.25}Cu_{0.25}Zn_{0.25}O with the rocksalt structure. This additional data will allow for a more comprehensive comparison between different compositions with the same structures.

Response 1: We thank Reviewer #2 for raising this suggestion. We synthesized Mg_{0.25}Ni_{0.25}Cu_{0.25}Zn_{0.25}O by the same method as the other high-entropy oxides in this work. The synthesized Mg_{0.25}Ni_{0.25}Cu_{0.25}Zn_{0.25}O has a single rock-salt phase (Fig. R#2-1). The prepared Mg_{0.25}Ni_{0.25}Cu_{0.25}Zn_{0.25}O electrode has a double-layer capacitance of 7.85 mF cm⁻² (Fig. R#2-2), bigger than 6.55 mF cm⁻² of Mg_{0.25}Co_{0.25}Ni_{0.25}Zn_{0.25}O and 3.81 mF cm⁻² of Mg_{0.20}Co_{0.20}Ni_{0.20}Cu_{0.20}Zn_{0.20}O. It means that Mg_{0.25}Ni_{0.25}Cu_{0.25}Zn_{0.25}O can be considered to have a largest electrochemical surface area. We compared the nitrate reduction activities of Mg_{0.25}Co_{0.25}Ni_{0.25}Zn_{0.25}O, Mg_{0.25}Ni_{0.25}Cu_{0.25}Zn_{0.25}O and Mg_{0.20}Co_{0.20}Ni_{0.20}Cu_{0.20}Zn_{0.20}O and observed an enhanced nitrate reduction activity on Mg_{0.20}Co_{0.20}Ni_{0.20}Cu_{0.20}Zn_{0.20}O (Fig. R#2-3). We have added the discussion in the revised manuscript (highlighted).

Fig. R#2-1. The XRD pattern of $\text{Mg}_{0.25}\text{Ni}_{0.25}\text{Cu}_{0.25}\text{Zn}_{0.25}\text{O}$. We have added it into the revised Supplementary Fig. 1.

Fig. R#2-2. The fitting curve and CV scans (inset) of the double-layer capacitance of $\text{Mg}_{0.20}\text{Ni}_{0.20}\text{Cu}_{0.20}\text{Zn}_{0.20}\text{O}$ at the scan rate of 100, 80, 60, 40, 20 and 10 mV s^{-1} in 1 M KOH. We have added it into the revised Supplementary Fig. 9.

Fig. R#2-3. LSV curves of nitrate reduction on $\text{Mg}_{0.25}\text{Co}_{0.25}\text{Ni}_{0.25}\text{Zn}_{0.25}\text{O}$ (RS-0), $\text{Mg}_{0.25}\text{Ni}_{0.25}\text{Cu}_{0.25}\text{Zn}_{0.25}\text{O}$ and $\text{Mg}_{0.20}\text{Co}_{0.20}\text{Ni}_{0.20}\text{Cu}_{0.20}\text{Zn}_{0.20}\text{O}$ (RS-20) in 1.0 M KOH + 0.1 M KNO_3 solution. We have added it into the revised Supplementary Fig. 10.

2. It is intriguing to explore the activity of Li-doped high entropy oxides at varying concentrations. Requesting information on the activity of a few Li-doped cases, in addition to the comparison between $\text{Mg}_{0.25}\text{Co}_{0.25}\text{Ni}_{0.25}\text{Zn}_{0.25}\text{O}$ and $\text{Li}_{0.20}\text{Mg}_{0.16}\text{Co}_{0.16}\text{Ni}_{0.16}\text{Cu}_{0.16}\text{Zn}_{0.16}\text{O}$, could provide insights into the role of Li and its concentration-dependent effects.

Response 2: We thank Reviewer #2 for raising this suggestion. We have supplemented two additional samples $\text{Li}_{0.10}\text{Mg}_{0.18}\text{Co}_{0.18}\text{Ni}_{0.18}\text{Cu}_{0.18}\text{Zn}_{0.18}\text{O}$ (Li-RS-18) and $\text{Li}_{0.30}\text{Mg}_{0.14}\text{Co}_{0.14}\text{Ni}_{0.14}\text{Cu}_{0.14}\text{Zn}_{0.14}\text{O}$ (Li-RS-14) for the Li-substituted MgCoNiCuZnO series from Li concentration 0, 10, 20 to 30 at%. The Li-substituted MgCoNiCuZnO series are all single rock-salt phase (Fig. R#2-4). Li-RS-18 and Li-RS-14 electrodes have the double-layer capacitance of 6.52 mF cm^{-2} and 1.92 mF cm^{-2} , respectively (Fig. R#2-5). The LSV curves show that the nitrate reduction activity decreases in the order of RS-20, Li-RS-18, Li-RS-16 and Li-RS-14 (Fig. R#2-6a), indicating that gradual Li substitution decreases nitrate reduction activity. After normalizing the double-layer capacitance and atomic ratio, although the Li-RS-14 has a highest current density at a large overpotential, RS-20 still has a better activity than those Li-substituted MgCoNiCuZnO from the onset potential. From the XANES results (Fig. R#2-7), the valence state of Co increases from RS-20, Li-RS-18 to Li-RS-16 with the Li introduction and has no obvious change from Li-RS-16 to Li-RS-14. For other metals, the valence state change caused by Li introduction is not obvious. From the EXAFS results (Fig. R#2-8), Li introduction has a most dramatic influence on Co. less Li introduction in Li-RS-18 can shorten the Co-O bond length, and more Li introduction can intensify the Co-O peak. We have added the relevant discussion in the revised manuscript (highlighted).

Fig. R#2-4. XRD patterns of $\text{Mg}_{0.20}\text{Co}_{0.20}\text{Ni}_{0.20}\text{Cu}_{0.20}\text{Zn}_{0.20}\text{O}$, $\text{Li}_{0.10}\text{Mg}_{0.18}\text{Co}_{0.18}\text{Ni}_{0.18}\text{Cu}_{0.18}\text{Zn}_{0.18}\text{O}$, $\text{Li}_{0.20}\text{Mg}_{0.16}\text{Co}_{0.16}\text{Ni}_{0.16}\text{Cu}_{0.16}\text{Zn}_{0.16}\text{O}$ and $\text{Li}_{0.30}\text{Mg}_{0.14}\text{Co}_{0.14}\text{Ni}_{0.14}\text{Cu}_{0.14}\text{Zn}_{0.14}\text{O}$. We have added it into the revised Supplementary Fig. 1.

Fig. R#2-5. The fitting curve and CV scans (inset) of the double-layer capacitance of (a) $\text{Li}_{0.10}\text{Mg}_{0.18}\text{Co}_{0.18}\text{Ni}_{0.18}\text{Cu}_{0.18}\text{Zn}_{0.18}\text{O}$ (Li-RS-18) and (b) $\text{Li}_{0.30}\text{Mg}_{0.14}\text{Co}_{0.14}\text{Ni}_{0.14}\text{Cu}_{0.14}\text{Zn}_{0.14}\text{O}$ (Li-RS-14) at the scan rate of 100, 80, 60, 40, 20 and 10 mV s^{-1} in 1 M KOH. We have added it into the revised Supplementary Fig. 9.

Fig. R#2-6. (a) LSV curves and (b) capacitance-normalized LSV curves, and atomic ratio (Cu and Co) and capacitance-normalized LSV curves (inset) of nitrate reduction on RS-0, Li-RS-18, Li-RS-16 and Li-RS-14 in 1.0 M KOH + 0.1 M KNO₃. We have added it into the revised Supplementary Fig. 7.

Fig. R#2-7. XANES of (a) Co, (b) Ni, (c) Cu and (d) Zn K-edges of Mg_{0.20}Co_{0.20}Ni_{0.20}Cu_{0.20}Zn_{0.20}O (RS-20), Li_{0.10}Mg_{0.18}Co_{0.18}Ni_{0.18}Cu_{0.18}Zn_{0.18}O (Li-RS-18), Li_{0.20}Mg_{0.16}Co_{0.16}Ni_{0.16}Cu_{0.16}Zn_{0.16}O (Li-RS-16), Li_{0.30}Mg_{0.14}Co_{0.14}Ni_{0.14}Cu_{0.14}Zn_{0.14}O (Li-RS-14) and the reference samples. We have added it into the revised Supplementary Fig. 3.

Fig. R#2-8. EXAFS of (a) Co, (b) Ni, (c) Cu and (d) Zn K-edges of $\text{Mg}_{0.20}\text{Co}_{0.20}\text{Ni}_{0.20}\text{Cu}_{0.20}\text{Zn}_{0.20}\text{O}$ (RS-20), $\text{Li}_{0.10}\text{Mg}_{0.18}\text{Co}_{0.18}\text{Ni}_{0.18}\text{Cu}_{0.18}\text{Zn}_{0.18}\text{O}$ (Li-RS-18), $\text{Li}_{0.20}\text{Mg}_{0.16}\text{Co}_{0.16}\text{Ni}_{0.16}\text{Cu}_{0.16}\text{Zn}_{0.16}\text{O}$ (Li-RS-16) and $\text{Li}_{0.30}\text{Mg}_{0.14}\text{Co}_{0.14}\text{Ni}_{0.14}\text{Cu}_{0.14}\text{Zn}_{0.14}\text{O}$ (Li-RS-14). We have added it into the revised Supplementary Fig. 2.

3. Although the authors have presented edx line scans demonstrating minimal changes on the surface during electrolysis, it would be valuable to provide surface information prior to the electrolysis process. This data would contribute to a better understanding of the initial surface state and facilitate a more comprehensive analysis of the electrochemical processes occurring during nitrate reduction.

Response 3: We thank Reviewer #2 for raising this suggestion. We carried out the EDX-mapping and EDX-line scans of $\text{Mg}_{0.20}\text{Co}_{0.20}\text{Ni}_{0.20}\text{Cu}_{0.20}\text{Zn}_{0.20}\text{O}$ (RS-20, Fig. R#2-9, 10) and $\text{Li}_{0.20}\text{Mg}_{0.16}\text{Co}_{0.16}\text{Ni}_{0.16}\text{Cu}_{0.16}\text{Zn}_{0.16}\text{O}$ (Li-RS-16, Fig. R#2-11, 12) prior to the electrolysis process. The difference in the elemental distribution from the bulk to the surface is hardly observed.

Fig. R#2-9. TEM-EDX mapping of RS-20 prior to nitrate reduction. We have added it into the revised Supplementary Fig. 18.

Fig. R#2-10. TEM-EDX line scans of RS-20 prior to nitrate reduction, (a-c) indicating the different regions. We have added it into the revised Supplementary Fig. 19.

Fig. R#2-11. TEM-EDX mapping of Li-RS-16 prior to the electrolysis. We have added it into the revised Supplementary Fig. 23.

Fig. R#2-12. TEM-EDX line scans of Li-RS-16 prior to the electrolysis, (a-c) indicating the different regions. We have added it into the revised Supplementary Fig. 24.

4. Expanding the discussion on the relationship between spin state and nitrate reduction would enhance the understanding of the proposed mechanism. Elaborate on how the spin state affects the catalytic activity and elaborate on any previous studies or theoretical frameworks that support the role of spin state in nitrate reduction processes.

Response 4: We thank Reviewer #2 for this suggestion. We have added it into the revised manuscript (highlighted) as below.

“Our results are also consistent with the recent theoretical findings by Chorkendorff, Nørskov and Wang (*Cao et al., Nat. Commun., 2022, 13, 2382; Cao and Nørskov, ACS Catal. 2023, 13, 3456-3462; Xu et al., J. Am. Chem. Soc. 2022, 144, 23089-23095*) on spin promoted ammonia synthesis from N₂ in Haber-Bosch process. The nitrate reduction to ammonia is a multi-step process, in which there is no N-N dissociation that occurs in Haber-Bosch process. The activity and selectivity trends of nitrate reduction on metals can be described by the adsorption strengths of atomic O and N. Due to the change of adsorbate from O to N and the intermediates during nitrate reduction, the adsorbate linear scaling relationships limit the potential maximum activity for single-site catalysts. In the Cu and Co combination, Cu favors reducing nitrate to nitrite and Co favors the selective reduction of N* to NH₃ due to strong nitrite binding (*Carvalho et al., Role of Electronic Structure on Nitrate Reduction to Ammonium: A Periodic Journey, J. Am. Chem. Soc. 2022, 144, 14809-14818*). The combination of Cu and Co breaks these scaling relations and reach the optimum point in the volcano plots and enhance the ammonia generation (*Liu et al., Activity and Selectivity Trends in Electrocatalytic Nitrate Reduction on Transition Metals, ACS Catal. 2019, 9, 7052-7064*). This enhancement from Cu and Co towards ammonia generation are reported in the alkaline (Wu et al., *ACS Sustainable Chem. Eng. 2022, 10, 14539–14548; Fang et al., Nat. Commun., 2022, 13, 7899*) and neutral electrolyte (*Jeon et al., J. Phys. Chem. C 2022, 126, 6982-6989; Liu et al., Chem. Eng. J., 2023, 466, 143134*). However, in Haber-Bosch process, a gradually increasing adsorption energy make the rate determining step of the reduction of N₂ to NH₃ shift from N₂ dissociation to NH_x hydrogenation, and too strong nitrogen binding energy limits the hydrogenation of NH_x (*Xu et al., J. Am. Chem. Soc. 2022, 144, 23089-23095*). Recent works by Chorkendorff, Nørskov and Wang demonstrate the intermediates N adsorption depends on the spin state (*Cao et al., Nat. Commun., 2022, 13, 2382; Cao and Nørskov, ACS Catal. 2023, 13, 3456-3462; Xu et al., J. Am. Chem. Soc. 2022, 144, 23089-23095*). Take Co as an example, the spin-polarized Co has a weaker adsorption toward to N and N-N transition state than non-spin-polarized Co (*Nat. Commun., 2022, 13, 2382; ACS Catal. 2023, 13, 3456-3462*). It means that the high spin of Co lifts the limit of the hydrogenation of NH_x and enhances the ammonia generation. Thus, the recent theoretical findings on spin promoted ammonia synthesis in Haber-Bosch process also support our experimental observation that the higher spin state of Co facilitates the ammonia generation in the process of electrochemical nitrate reduction and our calculation results that higher spin state of Co reduces the

barrier from NH_2 to NH_3 . Our results can fall well into the magnetic enhancement effect and be fitted into the spin promoted ammonia synthesis picture.”

5. Drawing a comparison between high entropy oxides and single atoms in terms of the feature of isolated homo-cations would be insightful. Discuss the similarities and differences between these two systems, highlighting their respective advantages and potential applications in catalysis.

Response 5: We thank Reviewer #2 for this suggestion. We added the discussion in the revised manuscript (highlighted) as below.

“Because these oxides have medium/high entropies, similar cation species like Ni and even intercalated Mg^{2+} and Zn^{2+} are well dispersed, which promotes the single-atom catalyst (SAC) like configurations, impeding dinitrogen formation and hydrogen evolution to enhance ammonia selectivity (*Chen et al. Nat. Energy, 2020, 5, 605-613*). It is reported that the selectivity diverges towards NH_3 and N_2 from the intermediates NO (*Wu et al. Nat. Commun. 2021, 12, 2870*). For better ammonia generation, inhibiting N-N coupling on the catalyst surface is necessary. The high-entropy oxides have the dispersed or isolated homo-cations, which could play a similar role work for inhibiting the N-N coupling like single-atom catalysts (SACs). SACs usually possess high selectivity toward specific products and the maximum atom utilization efficiency and quantum size effects (*Zhang et al., Adv. Energy Mater. 2018, 8, 1701343*). In SACs, the transition metal atoms are dispersed, which minimizes the N-N coupling possibility from two mono-nitrogen moieties on adjacent sites, which could inhibit N_2 formation and in turn promote the NH_3 selectivity. (*Wu et al., Nat. Commun., 2021, 12, 2870; Murphy et al., ACS Catal. 2022, 12, 6651–6662*) Compared to SACs, the high-entropy oxides have abundant catalytic active sites and high-temperature stability, which is helpful to break scaling relation limitations of adsorption energies. They are also treated as potential next-generation catalysts (*Pan et al., Chem. Eng. J., 2023, 451, 138659*).”

6. Comparing and discussing the stability of high entropy oxides and monoxides in terms of surface construction would provide valuable insights. Elaborate on the stability of high entropy oxides under the conditions of nitrate reduction and compare it to the stability of monoxides.

Response 6: We thank Reviewer #2 for raising this issue. We discuss the stability of these oxides from our XANES/EXAFS, HAADF-TEM EELS results prior to and after nitrate reduction combined with the reported Gibbs free energy, standard reduction potential, Pourbaix diagrams and valence state change / surface reconstruction in the literature. We added the relevant discussion into the revised manuscript (highlighted, discussion on Fig. 3) and on **Oxide stability** into the revised Supplementary Information.

The sample preparation for XANES and EXAFS of RS-20 and Li-RS-16 after nitrate reduction: Considering the oxide adhesion to the carbon paper and oxide loading mass for synchrotron measurement, the ink was prepared by sonicating the oxide and ECP-600JD with mass ratio 4 : 1 in a mixed solution (volume ratio, deionized water : isopropanol : 5 wt% Nafion solution = 3 : 1 : 0.1344) until the ink became homogeneous. The ink was dipped onto both sides of Toray 090 carbon paper and each side dimensions are $7 \times 7 \text{ mm}^2$. The total oxide mass loading is 30-40 mg cm^{-2} .

The sample preparation for TEM-EELS of RS-20 and Li-RS-16 after nitrate reduction: For observing the change of the surface or bulk and decrease the disturbance from carbon in TEM observance, we used nickel foam to replace carbon as the current collector and conductive network. We imbedded less amount oxide powders into the nickel foam then press them for a closer contact between oxide powders and nickel foam. After nitrate reduction, sonicating the imbedded nickel form to obtain the oxide particles for TEM observation.

The valence state and structure after nitrate reduction are revealed by XANES and EXAFS spectra (Fig. R#2-13). XANES results show that the Co/Ni/Cu K-edges shift to the lower energy direction, indicating the decrease of the Co, Cu and Ni valence states after nitrate reduction. The decrease of EXAFS peak intensity after nitrate reduction demonstrates that the structure becomes less ordered. For Co, according to the fitting of valence states and energies in Fig. R#2-14, the valence states of Co in RS-20 and Li-RS-16 decreases from 1.94 to 1.67 and 2.39 to 2.14, respectively. In Fig. R#2-13b, no metallic Co is observed by comparing the Co EXAFS results between our catalysts after nitrate reduction and Co foil. When similar analyses are applied to Ni and Cu, the slight reduction can be observed, and no metallic Ni/Cu signal can be found (Fig. R#2-13c-f). In addition, the Cu_2O pre-edge at 8983.7 eV in XANES is not observed after nitrate reduction (Fig. R#2-13e). It indicates that even though Cu_2O and Cu are formed after nitrate reduction, the ratio was very low.

Fig. R#2-13. XANES of (a) Co, (c) Ni and (e) Cu K-edges and EXAFS of (b) Co, (d) Cu and (f) Ni K-edges of $\text{Mg}_{0.20}\text{Co}_{0.20}\text{Ni}_{0.20}\text{Cu}_{0.20}\text{Zn}_{0.20}\text{O}$ (RS-20) and $\text{Li}_{0.20}\text{Mg}_{0.16}\text{Co}_{0.16}\text{Ni}_{0.16}\text{Cu}_{0.16}\text{Zn}_{0.16}\text{O}$ (Li-RS-16) before and after nitrate reduction at -0.35 V for 30 min in 1 M KOH + 0.1 M KNO_3 . Standard metals and metal oxides for reference data are attached in the figure. We have added it into the revised manuscript Fig. 3.

Fig. R#2-14. The estimation of Co valence states for the samples from the linear fitting of valence state and energy ($\mu\chi = 0.5$) of CoO and Co₃O₄ XANES. We have added it into the revised Supplementary Fig. 4.

We used EELS to probe the electronic states of Co and Ni and Cu at different locations of the bulk and surface before and after nitrate reduction (Fig. R#2-15-18). Statistical results are given in Fig. R#2-19. The change of Co L₃/L₂ intensity ratio can describe the change in Co valence states (Wang et al., *Micron*, 2000, 31, 571-580). Smaller ratios represent higher valence states. After nitrate reduction, Co L₃/L₂ ratios in RS-20 and Li-RS-16 increase at the surface and bulk, with the ratios at the surface bigger than the bulk. It is consistent with XANES results that Co can be reduced after reduction (Fig. 3). The Co valence state trend of RS-20 and Li-RS-16 from EELS measurement is agreement with our XANES results. Moreover, the bigger L₃/L₂ ratio at the surface indicates a bigger reduction degree. For Ni, the intensity between Ni L₃ and L₂ peaks can be used to evaluate the Ni valence state, and a higher intensity after the L₃ peak represents a lower valence state (Vila' et al., *Cell Reports Physical Science*, 2020, 1, 100188). For Cu, the location of the Cu L₃ peak can be used to evaluate the Cu valence state, and the peak location in the higher energy means a lower valence state (Laffont et al., *Micron*, 2006, 37, 459-464). In our EELS results, an obvious reduction of Ni and Cu can be observed, it can be explained through combining with XANES results that the reduction of Ni and Cu is slight (Fig. R#2-16, 18). We have added the relevant discussion like above into the revised Supplementary Information following Supplementary Fig. 29.

Fig. R#2-15. HAADF-STEM images for EELS measurements of RS-20 prior to and after nitrate reduction at -0.35 V for 30 min in 1 M KOH + 0.1 M KNO_3 . We have added it into the revised Supplementary Fig. 25.

Fig. R#2-16. Co, Ni and Cu EELS measurements for the bulk and surface of RS-20 prior to and after nitrate reduction at -0.35 V for 30 min in 1 M KOH + 0.1 M KNO_3 : (a) bulk prior to nitrate reduction, (b) surface prior to nitrate reduction, (c) bulk after nitrate reduction and (d) surface after nitrate reduction. All the data are aligned with oxygen peaks in Fig. 2f. We have added it into the revised Supplementary Fig. 26.

Fig. R#2-17. HAADF-STEM images for EELS measurements of Li-RS-16 prior to and after nitrate reduction at -0.35 V for 30 min in 1 M KOH + 0.1 M KNO₃. We have added it into the revised Supplementary Fig. 27.

Fig. R#2-18. Co, Ni and Cu EELS measurements for the bulk and surface of Li-RS-16 prior to and after nitrate reduction at -0.35 V for 30 min in 1 M KOH + 0.1 M KNO₃: (a) bulk prior to nitrate reduction, (b) surface prior to nitrate reduction, (c) bulk after nitrate reduction and (d) surface after nitrate reduction. All the data are aligned with oxygen peaks in Fig. 2f. We have added it into the revised Supplementary Fig. 28.

Fig. R#2-19. Co L₃/L₂ ratios from EELS measurements for the bulk and surface of RS-20 and Li-RS-16 prior to and after nitrate reduction at -0.35 V for 30 min in 1 M KOH + 0.1 M KNO₃. We have added it into the revised Supplementary Fig. 29.

Table R#2-1 shows that these individual monoxides have different Gibbs free energies. From the energy aspect, we can find the stability trend MgO > ZnO > NiO > CoO > CuO. It suggests Cu and Co are more susceptible to surface reduction than MgO. In addition, Calle-Vallejo et al. found the ΔG has a trend CuO > NiO > CoO, based on the reaction $M + H_2O(l) \rightarrow MO + 2(H^+ + e^-)$, indicating that the stability has trend CoO > NiO > CuO (*Calle-Vallejo et al., ACS Catal. 2015, 5, 869–873*).

Table R#2-1. ΔH^0 and S^0 and Gibbs free energy

	$\Delta H^0 / \text{kJ mol}^{-1}$	$S^0 / \text{J mol}^{-1} \text{K}^{-1}$	$\Delta G^0 / \text{kJ mol}^{-1}$
MgO	-601.60	26.95	-609.64
CoO	-237.74	52.85	-253.50
NiO	-246.60	38.58	-258.10
CuO	-156.06	42.59	-168.76
ZnO	-350.46	43.65	-363.47

ΔH^0 and S^0 are the standard enthalpy of formation at 298.15 K and the entropy at 298.15 K, respectively.

The Gibbs free energy is calculated according to $\Delta G = \Delta H - T\Delta S$. The values of ΔH^0 and S^0 for thermodynamics come from the database and website (*Haynes et al., Handbook of Chemistry and Physics 97th Edition, CRC Press, 2016-2017; Archer, J. Phys. Chem. Ref. Data, 1999, 28, 1485–1507; and <https://webbook.nist.gov/chemistry/>*).

Table R#2-2 shows the standard reduction potential from cations to metals. The standard reduction potentials of Cu, Co and Ni cations are more positive than that of Mg and Zn cations. It suggests that Cu, Co and Ni cations are easier to be reduced than Mg and Zn cations.

Table R#2-2. The reduction reaction and standard reduction potential

Reaction	E^0 / V
$\text{Mg}^{2+} + 2e^- \rightleftharpoons \text{Mg}$	-2.372
$\text{Mg}(\text{OH})_2 + 2e^- \rightleftharpoons \text{Mg} + 2\text{OH}^-$	-2.690
$\text{Co}^{2+} + 2e^- \rightleftharpoons \text{Co}$	-0.28

$\text{Co(OH)}_2 + 2\text{e}^- \rightleftharpoons \text{Co} + 2\text{OH}^-$	-0.73
$\text{Ni}^{2+} + 2\text{e}^- \rightleftharpoons \text{Ni}$	-0.257
$\text{Ni(OH)}_2 + 2\text{e}^- \rightleftharpoons \text{Ni} + 2\text{OH}^-$	-0.72
$\text{Cu}^{2+} + 2\text{e}^- \rightleftharpoons \text{Cu}$	0.3419
$\text{Cu(OH)}_2 + 2\text{e}^- \rightleftharpoons \text{Cu} + 2\text{OH}^-$	-0.222
$2\text{Cu(OH)}_2 + 2\text{e}^- \rightleftharpoons \text{Cu}_2\text{O} + 2\text{OH}^- + \text{H}_2\text{O}$	-0.080
$\text{Cu}_2\text{O} + \text{H}_2\text{O} + 2\text{e}^- \rightleftharpoons 2\text{Cu} + 2\text{OH}^-$	-0.360
$\text{Zn}^{2+} + 2\text{e}^- \rightleftharpoons \text{Zn}$	-0.7618
$\text{Zn(OH)}_2 + 2\text{e}^- \rightleftharpoons \text{Zn} + 2\text{OH}^-$	-1.249
$\text{ZnO} + \text{H}_2\text{O} + 2\text{e}^- \rightleftharpoons \text{Zn} + 2\text{OH}^-$	-1.260

E^0 represents the standard reduction potential and the value is versus the standard hydrogen electrode.

The data in the table refer to the database (*Haynes et al., Handbook of Chemistry and Physics 97th Edition, CRC Press, 2016-2017*).

Moreover, we also used the Pourbaix diagrams to compare the potential stability window (*Hochfilzer et al., ACS Energy Lett. 2023, 8, 1607-1612*). At the condition of pH = 14 and -0.35 V vs. RHE, Mg exists in Mg(OH)_2 (*Pesterfield et al., J. Chem. Educ. 2012, 89, 891-899*); Co in HCoO_2^- (*Garcia et al., J. Power Sources, 2008, 185, 549-553*) or metallic Co (*Powell et al., J. Chem. Educ. 1987, 64, 2, 165*); Ni in metallic Ni (*Huang et al., J. Phys. Chem. C 2017, 121, 9782-9789*); Cu in metallic Cu (*Protopopoff and Marcus, Electrochim. Acta, 2005, 51, 408-417*); Zn in ZnO/Zn(OH)_2 (*Borchers et al., J. Power Sources, 2021, 484, 229309*). It also indicates that Co, Ni and Cu are easier to be reduced to metals. Particularly, we also list some recent works on the reduction of Cu/Co compounds during/after nitrate reduction (Table R#2-3).

Table R#2-3. Some cases of Cu/Co reduction during/after the nitrate reduction

Raw catalysts	Under or after reaction	Potential (RHE)	Electrolyte	ref
Cu ₂ O particles	Cu	<-0.6 V	0.1 M neutral phosphate buffer solution + 0.1 M KNO ₃	ACS Catal. 2023, 13, 7529-7537
	Cu and Cu ⁺	≥ -0.6 V		
Cu ₂ O cubes	Cu	-0.5 V	0.5 M Na ₂ SO ₄ + 50 mM NaNO ₃ + NaOH (pH = 10)	J. Hazard. Mater., 2022, 439, 129653
CuO nanowire	Cu ₂ O	-0.45 V	0.5 M Na ₂ SO ₄ solution + 200 ppm nitrate-N	Angew. Chem. Int. Ed. 2020, 59, 5350-5354
CuO nanobelt	Cu	-0.4 V	1 M KOH + 0.5 M NO ₃ ⁻	Energy Environ. Sci., 2021, 14, 4989
Cu-Co binary sulfides	Cu/CuO _x -Co/CoO	-0.175 V	0.1M KOH + 0.01M KNO ₃	Nat. Commun., 2022, 13, 1129
CoO _x	Lower valence Co	-0.3 V	0.1 M KOH + 0.1 M KNO ₃	ACS Catal. 2021, 11, 15135-15140

Point-by-point response to Reviewer #3's comments

The authors test two high entropy oxides, MgCoNiCuZnO and LiMgCoNiCuZn for nitrate reduction to ammonia. They also test an oxide without Cu which also has low activity. The oxide without Li has a higher activity, which the authors attribute to high spin Co compared to low spin Co, where the low spin decreases the activity. This is based on electronic structure measurements. The concept of controlling beyond the elemental composition is interesting, but it is not clear to me that this is definitively shown to be a reason for enhancement on these systems. In addition, the per site activity of the catalysts is unclear in relation to other catalysts reported in the literature. Therefore, although the concept of new ways to control and go beyond scaling relation limitations of adsorption energies is interesting, it is not clear whether this publication clearly shows this. In addition, there are some details that need to be included to understand the results, particularly the experimental conditions, which are known to significantly impact the performance for nitrate reduction.

Response 1: We thank Reviewer #3 for raising this issue. $\text{Mg}_{0.20}\text{Co}_{0.20}\text{Ni}_{0.20}\text{Cu}_{0.20}\text{Zn}_{0.20}\text{O}$ has a higher activity toward nitrate reduction than $\text{Mg}_{0.25}\text{Co}_{0.25}\text{Ni}_{0.25}\text{Zn}_{0.25}\text{O}$. The nitrate reduction to ammonia is a multi-step process. It is reported that Cu is favorable in reducing nitrate to nitrite and Co favorable in reducing nitrite to ammonia. Thus, it is the synergistic effect from the tandem reduction from Cu and Co toward nitrate that enables $\text{Mg}_{0.20}\text{Co}_{0.20}\text{Ni}_{0.20}\text{Cu}_{0.20}\text{Zn}_{0.20}\text{O}$'s high activity. For $\text{Mg}_{0.20}\text{Co}_{0.20}\text{Ni}_{0.20}\text{Cu}_{0.20}\text{Zn}_{0.20}\text{O}$ and $\text{Li}_{0.20}\text{Mg}_{0.16}\text{Co}_{0.16}\text{Ni}_{0.16}\text{Cu}_{0.16}\text{Zn}_{0.16}\text{O}$, despite the presence of synergistic effects due to the coexistence of Cu and Co, Li incorporation suppresses this activity. The major changes in the electronic structure after Li incorporation include the increase in cobalt valence state, the decrease in cobalt spin state and the increase of holes in oxygen. There is no change in Cu and other metal elements. From the crystal structure, the oxides with or without Li all have single rock-salt phase. Local distortion has been alleviated after Li incorporation, resulting in a lower Co valence state and a shorter Co-O distance. Therefore, we attributed this activity suppression to the change of Co electronic state by Li incorporation. The intermediates adsorption is an important factor that determine the catalytic activity. Take N_2 reduction to NH_3 for example, a gradually increasing adsorption energy makes the rate determining step of N_2 reduction to NH_3 shift from N_2 dissociation to NH_x hydrogenation (*J. Am. Chem. Soc.* 2022, 144, 23089–23095). The recent works indicate that the spin-polarized Co has a weaker adsorption toward to N and N-N transition state than non-spin-polarized Co (*Nat. Commun.*, 2022, 13, 2382; *ACS Catal.* 2023, 13, 3456–3462), which means that the spin-polarized Co is favorable in NH_x hydrogenation. During the nitrate reduction to ammonia, the process of deoxygenation and hydrogenation happen and there is no N-N dissociation. In our work, we believe that high spin state Co favors NH_x hydrogenation, which is also confirmed by our calculation results that free energy change $\Delta G(\text{NH}_2^* \rightarrow \text{NH}_3^*)$ is lesser on high spin state Co than low spin state Co, and thus facilitates the nitrate reduction to ammonia.

For the issue “the per site activity of the catalysts is unclear in relation to other catalysts reported in the literature.”, it is difficult to calculate the per site activity accurately based on the following reasons. Though the reaction from nitrate to ammonia can be written as $\text{NO}_3^- + 6\text{H}_2\text{O} + 8\text{e}^- \rightarrow \text{NH}_3 + 9\text{OH}^-$, it is a multi-step process with the various intermediates. Our catalysts are high-entropy oxides including various metal elements. Although we can confirm the most active sites are Cu and Co, we cannot exclude the contribution from other metal elements like Ni. If we only treated the Cu and Co (atomic ratio 1 : 1) as the active sites, because Cu favors the reduction from nitrate to nitrite and Co favors the reduction from nitrite to ammonia, the currents passed through Cu and Co are different to be obtained accurately. Moreover, in the potential region of nitrate reduction, no redox peak is observed when the nitrate is absent, and thus we cannot estimate the number of atoms participating in the reaction either. Considering the various metal elements in the high-entropy oxides, this scenario is more complicated. If we use the electrochemical double-layer capacitance to estimate the electrochemical surface area, we cannot give the ratio of the crystal facets exposed to the electrolyte because our oxides are irregular. It also applies to other catalysts. Thus, we did not calculate the per site activity of the catalysts and gave the comparison with other catalysts reported in the literature. We just compared the activity based on the current density normalized by geometric area or double-layer capacitance.

We can give the estimated per site activity of RS-20 based on our LSV test and estimated electrochemical double-layer capacitance. The details are as follows.

For RS-20 on glassy carbon electrode, the measured electrochemical double layer capacitance is 3.81 mF $\text{cm}_{\text{geo}}^{-2}$. Considering the capacity of an ideal oxide surface roughness factor $60 \mu\text{F cm}^{-2}$ (*Baydi et al. J. Solid State Chem., 1995, 116, 157-169; Silva et al., Electrochim. Acta, 2001, 395-403*), the surface area can be converted into $63.5 \text{ cm}^2 \text{ cm}_{\text{geo}}^{-2}$. From the (111) peak in RS-20 XRD pattern, we can get the lattice constant $a = 4.2421 \text{ \AA}$. The surface area equals to $3.53 \times 10^{16} \text{ a}^2 \text{ cm}_{\text{geo}}^{-2}$. The estimated number of Co and Cu on different surfaces can be found in Table R#3-1, and the corresponding LSV curves are shown in Fig. R#3-1.

Table R#3-1. The estimated number of metal atoms of RS-20 on the glassy carbon electrode

surface	Metal atom density	Metal atom number	Estimated (Cu+Co) atom number
(100)	$2 / a^2$	7.06×10^{16}	2.824×10^{16}
(110)	$1.414 / a^2$	4.99×10^{16}	1.996×10^{16}
(111)	$2.309 / a^2$	8.15×10^{16}	3.26×10^{16}

Fig. R#3-1. LSV curves of RS-20 on glassy carbon electrode in 1 M KOH + 0.1 M KNO₃. The current is normalized by the number of Cu and Co atoms.

For the comment “Therefore, although the concept of new ways to control and go beyond scaling relation limitations of adsorption energies is interesting, it is not clear whether this publication clearly shows this.” We have added the discussion in the revised manuscript (highlighted) as below.

“Our results are also consistent with the recent theoretical findings by Chorkendorff, Nørskov and Wang (*Cao et al., Nat. Commun., 2022, 13, 2382; Cao and Nørskov, ACS Catal. 2023, 13, 3456–3462; Xu et al., J. Am. Chem. Soc. 2022, 144, 23089–23095*) on spin promoted ammonia synthesis from N₂ in Haber-Bosch process. The nitrate reduction to ammonia is a multi-step process, where no N-N dissociation occurs. The activity and selectivity trends of nitrate reduction on metals can be described by the adsorption strengths of atomic O and N. Due to the change of adsorbate from O to N and the intermediates during nitrate reduction, for single-site catalysts, the adsorbate linear scaling relationships limit the maximum possible activity. In Cu and Co combination, Cu is favorable in reducing nitrate to nitrite and Co favorable in the selective reduction of N* to NH₃ due to strong nitrite binding (*Carvalho et al., Role of Electronic Structure on Nitrate Reduction to Ammonium: A Periodic Journey, J. Am. Chem. Soc. 2022, 144, 14809–14818*). The combination of Cu and Co breaks these scaling relations and reach the optimum point in the volcano plots and enhance the ammonia generation (*Liu et al., Activity and Selectivity Trends in Electrocatalytic Nitrate Reduction on Transition Metals, ACS Catal. 2019, 9, 7052–7064*). This enhancement from Cu and Co towards ammonia generation are reported in the alkaline (*Wu et al., ACS Sustainable Chem. Eng. 2022, 10, 14539–14548; Fang et al., Nat. Commun., 2022, 13:7899*) and neutral electrolyte (*Jeon et al., J. Phys. Chem. C 2022, 126, 6982–6989; Liu et al., Chem. Eng. J., 2023, 466, 143134*). However, in Haber-Bosch process, a gradually increasing

adsorption energy makes the rate determining step of the reduction of N_2 to NH_3 shift from N_2 dissociation to NH_x hydrogenation, and much stronger nitrogen binding energy limits the hydrogenation of NH_x (*J. Am. Chem. Soc.* 2022, 144, 23089–23095). Recent works by Chorkendorff, Nørskov and Wang demonstrates the intermediates N adsorption depends on the spin state (*Cao et al., Nat. Commun.*, 2022, 13, 2382; *Cao and Nørskov, ACS Catal.* 2023, 13, 3456–3462; *Xu et al., J. Am. Chem. Soc.* 2022, 144, 23089–23095). Take Co as example, the spin-polarized Co has a weaker adsorption toward to N and N-N transition state than non-spin-polarized Co (*Nat. Commun.*, 2022, 13, 2382; *ACS Catal.* 2023, 13, 3456–3462). It means that the high spin of Co facilitates the hydrogenation of NH_x and thus enhances the ammonia generation. Thus, the recent theoretical findings on spin promoted ammonia synthesis in Haber-Bosch process also supports our experimental observation that the higher spin state of Co facilitates the ammonia generation in the process of electrochemical nitrate reduction and our calculation results that higher spin state of Co reduces the barrier from NH_2 to NH_3 . Our results can fall well into the magnetic enhancement effect and be fitted into the spin promoted ammonia synthesis picture.”

An additional positive is the use of the flow cell to control mass transport, as there are only a few studies that have operated NO₃RR in a flow cell (ref 13 cited being one of them I believe). The use of this to control mass transport and operate in continuous mode is interesting, but as mentioned below brings into question some of the fundamental insights. Regardless, I did find this aspect to be novel in the NO₃RR literature and perhaps something that should be done more often. As it is a somewhat interesting component, I found the description of the cell and its operating conditions a bit confusing and it could be elaborated on.

Response 2: We thank Reviewer #3 to raise this issue.

In our work, we indeed found employing the flow cell can promote the ammonia generation and Faradaic efficiency. Because of the novelty in the usage of the flow cell for nitrate reduction (*Wang et al., J. Am. Chem. Soc. 2020, 142, 5702–5708; Guo et al., ACS Sustainable Chem. Eng. 2023, 11, 7882-7893*), we cited some relevant works and emphasized them in **Results-Flow cell test** in revised manuscript (highlighted).

We also elaborated on the **Methods-Flow cell test** in the revised manuscript (highlighted) as below. **“Flow cell test.** The electrochemical nitrate reduction reaction in a flow cell was conducted in a self-designed polyether ether ketone (PEEK) flow cell modified from the reference¹³. The flow cell consisted of the gas, catholyte and anolyte chambers. The gas and cathodic chambers were separated by a gas diffusion electrode (GDE) Sigracet 38BC (SGL Carbon). The catholyte and anolyte chambers were separated by an Fumasep FAA-3-PK-130 (Fumatech) anion-exchange membrane (AEM). The catalyst ink was prepared by sonicating the oxide and ECP-600JD with mass ratio 4 : 1 in a mixed solution (volume ratio, deionized water : isopropanol : 5 wt% Nafion solution = 3 : 1 : 0.1344) until the ink became homogeneous. The ink was air sprayed onto Sigracet 38BC (SGL Carbon) gas diffusion electrode on the single side. The area of the working electrode was $2 \times 2 \text{ cm}^2$, and the catalyst loading was $0.5 \text{ mg}_{\text{oxide}} \text{ cm}^{-2}$. Catalysts loaded Sigracet 38BC (SGL Carbon) gas diffusion electrode, a leak-free Ag/AgCl/3.4 M KCl electrode (0.204 V vs. standard hydrogen electrode, Innovative Instruments) and a Pt plate ($2 \times 2 \text{ cm}^2$) were employed as working, reference and counter electrodes, respectively. The thickness of the catholyte chamber was reduced to 4 mm only in our customized design to minimize the system resistance. The anolyte chamber dimensions are $2 \text{ cm} \times 2 \text{ cm} \times 1 \text{ cm}$. Prior to the electrolysis, Ar gas steam is used to purge the gas out of the electrolyte, 1 M KOH with various concentration (0.01, 0.05, 0.1 and 0.5 M) KNO₃ to avoid the possible oxygen reduction reaction. Then, 30 mL electrolyte was pumped into both the catholyte and anolyte chambers of the flow cell, respectively, with a flow rate of 5 mL min^{-1} by (LongerPump, BT300-2J). The catholyte (30 mL) and anolyte (30 mL) were recycled in both chambers. Meanwhile, the Ar was purged into the gas chamber with a flow rate of 50 mL min^{-1} . The solution resistances between working electrode and reference electrode are 0.595 Ohm (1 M KOH), 0.4546 Ohm (1 M KOH + 0.01 M KNO₃), 0.264 Ohm (1 M KOH + 0.05 M KNO₃), 0.254 Ohm (1 M KOH + 0.1 M KNO₃) and 0.382 Ohm (1 M KOH + 0.5 M KNO₃), respectively. The potential

was applied at 0.1, 0, -0.1, -0.2, -0.3 and -0.4 V vs. RHE for 30 min, respectively for ammonia generation.”

The authors probe the spin state in part based on spin-related work from Chorkendorff and Norskov recently for promoters of Haber Bosch process (magnetic enhancement and electrostatic). It is not completely clear to me from the manuscript whether the authors believe the effects of the spin state fall into the magnetic enhancement effects or electrostatic effects. A more detailed discussion of the recent paper by Chorkendorff and Norskov, and how the results here fit into this picture is needed. As it is written, it is hard to interpret without doing significant reading of other literature, some of which is quite recent.

Response 3: We appreciate Reviewer #3 for raising this issue, which gives us an insightful understanding towards this work. We believe that the effect of spin state in our work can fall into the magnetic enhancement effects and our results can be fitted into spin state-adsorption-activity picture, however, we did not find sufficient evidence to explain it through electrostatic effects.

In this work by Chorkendorff and Norskov (*Cao et al., A spin promotion effect in catalytic ammonia synthesis, Nat. Commun., 2022, 13, 2382*), authors considered that an enhancement of the rate in ammonia synthesis on Ru will primarily come from a lowering of the N-N transition state energy for N₂ dissociation for weak-bonding catalysts, which is explained by the electrostatic effects after the promotor Cs, K, Li, Ba, Ca and La introduction. For magnetic catalysts, like spin-polarized Co, an extra promotion effect is observed, directly proportional to the promoter-induced reduction in spin moment of the Co atoms. Compared with spin-polarized Co, the energies of adsorbed N and of the N₂ dissociation transition state are lower in non-spin polarized Co. They also pointed out the differences in adsorption arises from the d-band splitting, such that the average adsorption energy of spin-up and spin-down components is less negative than the non-spin-polarized version. Furthermore, Cao and Nørskov found that this spin effect on adsorption is general. The adsorption energy of N and the N₂ dissociation transition state decreases as the spin moment decreases. (*Cao and Nørskov, Spin Effects in Chemisorption and Catalysis, ACS Catal. 2023, 13, 3456–3462*) Besides, Xu et al. reported that the paramagnetic Co and Ni metals could have higher ammonia synthesis activity than their ferromagnetic counterparts. For Co, the ferromagnetic-paramagnetic transition makes the N binding energy become more negative bypassing the Sabatier optimal and cause the rate-limiting steps to shift from N₂ dissociation to NH_x hydrogenation. (*Xu et al., Toward Sabatier Optimal for Ammonia Synthesis with Paramagnetic Phase of Ferromagnetic Transition Metal Catalysts, J. Am. Chem. Soc. 2022, 144, 23089–23095*)

In our ammonia generation process, nitrogen source is NO₃⁻ not N₂. The pathway from NO₃⁻ to NH₃ is different from N₂ to NH₃. From N₂ to NH₃, it involves the N₂ adsorption, N-N dissociation, N hydrogenation and NH₃ dissociation. From NO₃⁻ to NH₃, the first step is NO₃⁻ adsorption, and the adsorbed atom is O not N, then deoxygenation occurs from NO₃⁻ to N not N-N bond dissociation. Cu is

commonly believed to be favorable in the reduction from the nitrate to nitrite. Co facilitates the reduction from nitrite to ammonia, which involves the NH_x hydrogenation. Based on the above statement, the high spin state of Co has a more positive N binding energy (less stable) than low spin state of Co. The high spin state of Co favors the hydrogenation of NH_x compared to the low spin state of Co. It leads to a better ammonia generation. This explanation is consistent with our calculation results that the step from NH_2 to NH_3 is easier on Cu-Co-doped MgO(111) surface structures than on Co-doped MgO(111) surface structures (Fig. 6 in the revised manuscript). Thus, we believe our results can fall into the magnetic enhancement effect and be fitted into the picture.

We have added the discussion in the revised manuscript same with **Response 1**.

On a per site basis how good are the catalysts compared to similar conditions (cations, anions in electrolyte, pH, potential, temperature all kept constant, nitrate concentration constant)? For example, in the sentence “Our ammonia generation performance is comparable to other works (Table S1), despite the differences in electrode preparation, applied potential, solution pH value, nitrate concentration, and normalizations of performance indicators.” I do not understand why this would be relevant, since it may just be a coincidence that the activity is the same, as all of those parameters are well-known to affect the ammonia generation.

Response 4: We thank Reviewer #3 for this question. Though we can compare our catalysts with others, we cannot do it accurately to per site. The primary reason is that in the high-entropy oxide, we cannot calculate the number of Cu and Co active sites exposed to the electrolyte, and meanwhile we cannot fully exclude the contribution from other metal cations. The possible surface reconstruction also affects the evaluation of the number of active sites. We gave our estimated activity on a per site basis for our samples and have given the explanation in **Response 1**.

According to Reviewer #3, we updated the Tables S1 in the revised Supplementary Table 4, which focuses the work of Cu- and Co-based materials in 0.1-1 M KOH, as below.

	Catalyst	electrolyte	NH ₃ FE	NH ₃ yield rate	potential	Year / ref
1	Mg _{0.20} Co _{0.20} Ni _{0.20} Cu _{0.20} Zn _{0.20} O	1 M KOH + 0.1 M NO ₃ ⁻ 1 M KOH + 0.5 M NO ₃ ⁻	99.3% 97.2%	5.05 mg mg _{cat} ⁻¹ h ⁻¹ 26.6 mg mg _{cat} ⁻¹ h ⁻¹	-0.2 V vs. RHE -0.4 V vs. RHE	This work
2	Ru-Cu nanowire	1 M KOH + 0.032 M NO ₃	96.0%	76.563 mg h ⁻¹ cm ⁻²	0.04 V vs. RHE	2022 ³⁰
3	Cu ₅₀ Co ₅₀ / nickel foam	1 M KOH + 0.1 M KNO ₃	~100%	4.8 mmol cm ⁻² h ⁻¹	-0.2 V vs. RHE	2022 ³¹
4	Co _{0.5} Cu _{0.5} / carbon fiber	1 M KOH + 0.05 M NO ₃ ⁻	>95%	176 mA cm ⁻²	-0.03 V vs. RHE	2022 ³²
5	Ar-plasma treated Cu ₃₀ Co ₇₀ / carbon paper	1 M KOH + 0.1 M NO ₃ ⁻	~80%	5.13 mg cm ⁻² h ⁻¹	-0.47 V vs. RHE	2022 ³³
6	Cu-Co binary sulfides evolved Cu/CuO _x -Co/CoO hybrids	0.1 M KOH + 0.01 M NO ₃ ⁻	93.3%	1.17 mmol cm ⁻² h ⁻¹	-0.175 V vs. RHE	2022 ²⁸
7	ZnCo ₂ O ₄ / carbon paper	0.1 M KOH + 0.1 M NO ₃ ⁻	95.4%	2.10 mg mg ⁻¹ h ⁻¹	-0.4 V vs. RHE	2022 ³⁴
8	CoO/N-doped carbon nanotube/graphite paper	0.1 M KOH + 0.1 M NO ₃ ⁻	93.8%	9.04 mg h ⁻¹ cm ⁻²	-0.6 V vs. RHE	2022 ³⁵
9	Ultrathin CoO _x nanosheets	0.1 M KOH + 0.1 M NO ₃ ⁻	93.4%	82.4 mg h ⁻¹ mg ⁻¹	-0.3 V vs. RHE	2021 ²⁹
10	Cu ₅₀ Ni ₅₀ / PTFE	1 M KOH + 0.1 M NO ₃ ⁻	99%	-	-0.15 V vs. RHE	2020 ³⁶
11	Cu single atom	0.1 M KOH + 0.1 M NO ₃ ⁻	84.7%	4.5 mg cm ⁻² h ⁻¹ / 12.5 mol g _{Cu} ⁻¹ h ⁻¹	-1.0 V vs. RHE	2022 ³⁷
12	Cu(100)-rich rugged Cu-nanobelt	1 M KOH + 0.1 M NO ₃	95.3%	650 mmol h ⁻¹ g _{cat} ⁻¹ h ⁻¹	-0.15 V vs. RHE	2021 ²⁷
13	Cu ₂₊₁ O/Ag- carbon cloth	0.1 M KOH + 0.01 M KNO ₃	85.03%	2.2 mg h ⁻¹ cm ⁻²	-0.74 V vs RHE	2023 ³⁸
14	Cu-doped Fe ₃ O ₄	0.1 M KOH + 0.1 M KNO ₃	~100%	179.55 mg h ⁻¹ mg _{cat} ⁻¹	-0.6 V vs RHE	2023 ³⁹
15	Cu-modified Ru/C	0.1 M NaOH + 0.1 M NaNO ₃	95%	23.7 μmol h ⁻¹ cm ⁻² 6.86 mmol h ⁻¹ mg _{Ru} ⁻¹	-0.1 V vs. RHE	2023 ⁴⁰
16	Cu@nickel foam	1 M KOH + 200 ppm nitrate-N	96.6%	0.252 mmol h ⁻¹ cm ⁻²	-0.23 V vs. RHE	2021 ⁴¹

In my opinion, it is very difficult to deconvolute the effects that contribute to the nitrate reduction activity and selectivity. In particular, having two samples makes it a challenge as there are multiple factors being varied. A stronger case of the spin state would be to have control of it over a series of values and show how the activity correlates, rather than only three data points (RS-0, RS-20 and Li-RS-16). The authors compare and imply that RS-0 and RS-20 are evidence of the spin state being a factor (RS-0 has no Cu, and RS-20 has higher spin state), but it is unclear how the Cu presence may affect this. Is there a way to tune the amount of Li so that the spin state of the Co can be systematically varied and plotted against the activity under controlled conditions?

Response 5: We thank Reviewer #3 for the question. We supplemented another three samples $\text{Mg}_{0.225}\text{Co}_{0.225}\text{Ni}_{0.225}\text{Cu}_{0.10}\text{Zn}_{0.225}\text{O}$ (RS-10), $\text{Li}_{0.10}\text{Mg}_{0.18}\text{Co}_{0.18}\text{Ni}_{0.18}\text{Cu}_{0.18}\text{Zn}_{0.18}\text{O}$ (Li-RS-18) and $\text{Li}_{0.30}\text{Mg}_{0.14}\text{Co}_{0.14}\text{Ni}_{0.14}\text{Cu}_{0.14}\text{Zn}_{0.14}\text{O}$ (Li-RS-14) in the series. The Cu ratio increases from 0, 10 to 20 in RS-0, RS-10 and RS-20, respectively. The Li ratio increases from 0, 10, 20 to 30 in RS-20, Li-RS-18, Li-RS-16 and Li-RS-14, respectively. The XRD patterns shows that all these samples are single rock-salt phase (Fig. R#3-1). Figure R#3-2 shows CV curves and current density-scan rate linear fittings for electrochemical double-layer capacitance of RS-10, Li-RS-18 and Li-RS-14. Fig. R#3-3a-3b shows increasing nitrate reduction current density as the Cu atomic ratios increase from 0, 10 to 20, indicating that the increase of Cu promotes the nitrate reduction. Fig. R#3-3c-3d shows a decreasing nitrate reduction current density normalized by the glassy carbon area when the Li ratios increases from 0, 10, 20 to 30. When the current density is normalized by the electrochemical double-layer capacitance, the current density of Li-RS-14 becomes higher than other samples at a large overpotential (Fig. R#3-3d). This could be caused by the smaller electrochemical double layer capacitance of Li-RS-14. When comparing the nitrate reduction by onset potentials, RS-20 still has a higher nitrate reduction activity.

From Co $2p_{3/2}$ XPS results, the Co^{2+} satellite peak exists in RS-0, RS-10 and RS-20. Because we did not observe an increasing Co^{2+} satellite/main ratio with the increase of Cu ratios from RS-0, RS-10 to RS-20, we removed the discussion on the increase of Co spin state with the presence of Cu. Co^{2+} satellite peak started to disappear with the introduction of Li (Fig. R#3-4). Only a trace amount of Co^{2+} satellite peak is observed in Li-RS-18. This suggests that the high spin state of Co cations decrease even disappear. From Cu $2p_{3/2}$ XPS results, the Cu^+ ratio increases with the increase of Cu from RS-0, RS-10 to RS-20 and then decreases with the introduction of Li from RS-20, Li-RS-18, Li-RS-16 to Li-RS-14. The Cu^+ could also be a factor that influence the nitrate reduction activity. Because the signal of Cu^+ was not observed from XANES, it could exist only in the surface. Li introduction decreases the Cu^+ ratio in the surface. Though we cannot give a series of gradient Co spin states and their corresponding nitrate reduction activity, it is sufficient to conclude that the Co high spin in Co-Cu is necessary for high nitrate reduction activity. We have added these results into the revised manuscript (highlighted) and revised Supplementary Information.

Fig. R#3-1. XRD patterns of $\text{Mg}_{0.25}\text{Co}_{0.25}\text{Ni}_{0.25}\text{Zn}_{0.25}\text{O}$ (RS-0), $\text{Mg}_{0.225}\text{Co}_{0.225}\text{Ni}_{0.225}\text{Cu}_{0.10}\text{Zn}_{0.225}\text{O}$ (RS-10), $\text{Mg}_{0.20}\text{Co}_{0.20}\text{Ni}_{0.20}\text{Cu}_{0.20}\text{Zn}_{0.20}\text{O}$ (RS-20), $\text{Li}_{0.10}\text{Mg}_{0.18}\text{Co}_{0.18}\text{Ni}_{0.18}\text{Cu}_{0.18}\text{Zn}_{0.18}\text{O}$ (Li-RS-18), $\text{Li}_{0.20}\text{Mg}_{0.16}\text{Co}_{0.16}\text{Ni}_{0.16}\text{Cu}_{0.16}\text{Zn}_{0.16}\text{O}$ (Li-RS-16) and $\text{Li}_{0.30}\text{Mg}_{0.14}\text{Co}_{0.14}\text{Ni}_{0.14}\text{Cu}_{0.14}\text{Zn}_{0.14}\text{O}$ (Li-RS-14).

We have added it into the revised Supplementary Fig. 1.

Fig. R#3-2. CV curves in 1 M KOH and current density-scan rate linear fittings to obtain electrochemical double-layer capacitances: (a) RS-10, (b) Li-RS-18 and (c) Li-RS-14. The geometry area of the working electrode (glassy carbon) was 0.19625 cm^2 , and the catalyst loading was $1 \text{ mg}_{\text{oxide}} \text{ cm}^{-2}$. We have added it into the revised Supplementary Fig. 9.

Fig. R#3-3. LSV curves of RS-0, RS-10, RS-20, Li-RS-18, Li-RS-16, and Li-RS-14. The increase of the Cu ratio in (a) and (b). The increase of the Li ratio in (c) and (d). The current density is normalized by glassy carbon's geometric area of 0.19625 cm^2 in (a) and (c). The current density is normalized by the electrochemical double-layer capacitance and Co + Cu atomic ratios (inset) in (b) and (d). Experimental condition: The mass loading of $1 \text{ mg}_{\text{oxide}} \text{ cm}^{-2}$ on the glassy carbon substrate (geometry area 0.19625 cm^2) for LSV tests in Ar-saturated 1 M KOH with or without 0.1 M KNO_3 at a scan rate of 10 mV s^{-1} with 85% IR correction. We have added it into the revised Supplementary Fig. 7.

Fig. R#3-4. XANES of (a) Co, (b) Ni, (c) Cu and (d) Zn K-edges of RS-0, RS-10, RS-20, Li-RS-18, Li-RS-16, Li-RS-14 and the reference samples. We have added it into the revised Supplementary Fig.

Fig. R#3-5. EXAFS of (a) Co, (b) Ni, (c) Cu and (d) Zn K-edges of RS-0, RS-10, RS-20, Li-RS-18, Li-RS-16 and Li-RS-14. We have added it into the revised Supplementary Fig. 2.

Fig. R#3-6. (a) Co and (b) Ni XPS of RS-0, RS-10, RS-20, Li-RS-18, Li-RS-16 and Li-RS-14. We have added it into the revised Supplementary Fig. 5.

Fig. R#3-7. (a) Cu XPS and (b) EELS measurement of oxygen of RS-0, RS-10, RS-20, Li-RS-18, Li-RS-16 and Li-RS-14. We have added it into the revised Supplementary Fig. 6.

Can the authors compare to Cu and Co (and CuCo alloy) controls under the same conditions to show whether there is an enhancement? Ideally with controlled mass transport to avoid any artifacts and with surface area or site normalization. It is hard to understand what is meant by “high-entropy oxide RS-20 is a decent candidate”.

Response 6: We thank Reviewer #3 for giving us these suggestions. In order to address this issue, we first list some published works that compare the nitrate reduction activities of Cu, Co and CuCo. Secondly, we synthesized these Cu, Co and CuCo and then compared the nitrate reduction activity by employing H-cell and flow cell. We deleted this statement “high-entropy oxide RS-20 is a decent candidate” in the revised manuscript.

Firstly, the activity of Cu, Co and CuCo alloy has been reported by some groups. For example, Wu et al. published “Boosting electrocatalytic nitrate-to-ammonia conversion via plasma enhanced CuCo alloy–substrate interaction” in *ACS Sustainable Chem. Eng.* 2022, 10, 14539–14548. They found CuCo alloy has a higher activity than Cu and Co toward the reduction of nitrate to ammonia (electrolyte: 1 M KOH + 0.1 M KNO₃). This high activity of the CuCo alloy was attributed to weakening the strong adsorption capacity of Cu after introducing Co and lowering the energy barrier by shifting the d-bands. Fang et al. published “Ampere-level current density ammonia electrochemical synthesis using CuCo nanosheets simulating nitrite reductase bifunctional nature” in *Nat. Commun.*, 2022, 13:7899, Fang et al. electrodeposited a series of bimetallic CuCo materials on Ni foam for ammonia generation, and the optimized composition is Co_{0.5}Cu_{0.5} for nitrate reduction and ammonia production faradaic efficiency (electrolyte: 1 M KOH + 0.1 M KNO₃). They proposed a rational *H adsorption on the surface is the key to controlling the intermediates adsorption for an excellent performance. The Co introduction reduce the barrier of *NO₃ adsorption on Cu, and hydrogenation energy of *NO_x is also reduced due to facile *H adsorption on Co than Cu. Jeon et al. published “Cobalt-copper nanoparticles on three-dimensional substrate for efficient ammonia synthesis via electrocatalytic nitrate reduction” in *J. Phys. Chem. C* 2022, 126, 6982–6989. A series of Co_{1-x}Cu_x nanoparticles were investigated for ammonia generation, and the optimized composition is Co_{0.5}Cu_{0.5} for nitrate reduction and ammonia production faradaic efficiency (Electrolyte: 1 M KOH + 0.05 M KNO₃). Liu et al. prepared the Cu_xCo_y conductive metal-organic frameworks found Cu₁Co₁ exhibited a better ammonia generation than other compositions (Electrolyte: 0.5 M Na₂SO₄ + 0.1 M NaNO₃) (*Liu et al., Cu/Co bimetallic conductive MOFs: Electronic modulation for enhanced nitrate reduction to ammonia, Chem. Eng. J.*, 2023, 466, 143134).

Secondly, we prepared the Cu, Co and CuCo according to the procedure below. The XRD patterns are shown in Fig. R#3-8. The synthesized method is one-step reduction of metal ions by NaBH₄ (*Wang et al., Journal of Alloys and Compounds*, 2015, 651, 382-388; *Ghafar et al., Journal of The*

Electrochemical Society, 2022, 169, 096507). The detailed procedure is as follow. Take $\text{Cu}_{0.5}\text{Co}_{0.5}$ as an example.

Solution A: 5 mmol $\text{CuCl}_2 \cdot 2\text{H}_2\text{O}$ and 5 mmol $\text{CoCl}_2 \cdot 6\text{H}_2\text{O}$ were dissolved in 100 mL deionized water with stirring water for 10 min.

Solution B: 100 mmol NaBH_4 was added into 50 mL of 0.01 M NaOH solution with stirring for 10 min.

The freshly prepared **Solution B** was added into **Solution A** drop by drop in an ice bath. The mixed solution was kept stirring in the ice bath. The obtained black products were collected when the gas generation stop and then washed with deionized water and absolute ethanol four times successively by centrifugation. The precipitation was dried at 60°C for 10 h in a vacuum oven. For Cu and Co synthesis, 10 mmol $\text{CuCl}_2 \cdot 2\text{H}_2\text{O}$ and 10 mmol $\text{CoCl}_2 \cdot 6\text{H}_2\text{O}$ were used, respectively.

Fig. R#3-8. XRD patterns of the synthesized Cu, Co and $\text{Cu}_{0.5}\text{Co}_{0.5}$. The standard XRD patterns Co (JCPDS 00-015-0806, cubic, Fm3m), Cu (JCPDS 00-004-0836, cubic, Fm3m) and Cu_2O (JCPDS 00-005-0667, cubic, Pn3m) are shown in drop lines. The synthesized $\text{Cu}_{0.5}\text{Co}_{0.5}$ has a cubic structure, and the synthesized Cu has Cu_2O impurity due to Cu oxidation, which is reported in the literature (*Glavee et al., Langmuir 1994,10, 4726-4730*). No metallic peak can be observed, which could be due to the formation of Co_xB (*Ghafar et al., Journal of The Electrochemical Society, 2022, 169, 096507*).

We have also obtained the electrochemical double-layer capacitance and compared the nitrate reduction activity of Cu, Co and CuCo in Fig. R#3-9, 10. We found the nitrate reduction trend $\text{CuCo} > \text{Cu} > \text{Co}$ in Fig. R#3-10a. We noticed that normalizing the current density by the electrochemical double-layer capacitance could not be suitable for comparing the activity due to the instability of these metal particles in alkaline, in particular, the addition of Co increases the double-layer capacitance to a large degree. It can cause the change of nitrate reduction activity trend (Fig. R#3-10). We only use the geometric area normalized current density to compare the activity of CuCo and RS-20 in H-cell and flow cell (Fig. R#3-11). CuCo has a higher nitrate reduction activity than RS-20 in H-cell (Fig. R#3-11a). When employing the flow cell, the activity of CuCo and RS-20 are very close. It suggests that many details need to be optimized when using flow cell to compare the activity of different materials, as mentioned in the work by Tarpeh's group (*Guo et al. ACS Sustainable Chem. Eng. 2023, 11, 7882-7893*).

Fig. R#3-9. CV curves and current density-scan rate linear fittings for electrochemical double-layer capacitance in 1 M KOH: (a) Cu, (b) Co and (c) CuCo.

Fig. R#3-10. LSV curves of Cu, Co and CuCo in 1 M KOH + 0.1 M KNO_3 solution: (a) current density normalized by glassy carbon geometry area 0.19625 cm^2 and (b) current density normalized by electrochemical double-layer capacitance.

Fig. R#3-11. LSV curves of CuCo and RS-20 in 1 M KOH + 0.1 M KNO_3 solution. Setup: (a) H-cell and (b) flow cell. The current density is normalized by the geometric area. The H-cell and flow cell experimental condition can be found in the revised manuscript.

For the characterization (XANES, etc), how relevant is it to the structure under reaction conditions (i.e. after reconstruction) and how relevant is the bulk structure from XANES/EXAFS to the surface structure that is important for electrocatalysis?

Response 7: We thank Reviewer #3 for pointing out this issue. To resolve this issue, we measured the XANES and EXAFS of $\text{Mg}_{0.20}\text{Co}_{0.20}\text{Ni}_{0.20}\text{Cu}_{0.20}\text{Zn}_{0.20}\text{O}$ (RS-20) and $\text{Li}_{0.20}\text{Mg}_{0.16}\text{Co}_{0.16}\text{Ni}_{0.16}\text{Cu}_{0.16}\text{Zn}_{0.16}\text{O}$ (Li-RS-16) prior to and after nitrate reduction for the bulk information and used HADDF-STEM EELS to compare the difference between the bulk and surface before and after nitrate reduction.

The sample preparation for XANES and EXAFS of RS-20 and Li-RS-16 after nitrate reduction: Considering the oxide adhesion to the carbon paper and oxide loading mass for synchrotron measurement, the ink was prepared by sonicating the oxide and ECP-600JD with mass ratio 4 : 1 in a mixed solution (volume ratio, deionized water : isopropanol : 5 wt% Nafion solution = 3 : 1 : 0.1344) until the ink became homogeneous. The ink was dipped onto both sides of Toray 090 carbon paper and each side dimensions are $7 \times 7 \text{ mm}^2$. The total oxide mass loading is 30-40 mg cm^{-2} .

The sample preparation for TEM-EELS of RS-20 and Li-RS-16 after nitrate reduction: For observing the change of the surface or bulk and decrease the disturbance from carbon in TEM observance, we used nickel foam to replace carbon as the current collector and conductive network. We imbedded less amount oxide powders into the nickel foam then press them for a closer contact between oxide powders and nickel foam. After nitrate reduction, sonicating the imbedded nickel form to obtain the oxide particles for TEM observation.

The valence state and structure after nitrate reduction are revealed by XANES and EXAFS spectra (Fig. R#3-12). XANES results show that the Co/Ni/Cu K-edges shift to the lower energy direction, indicating the decrease of the Co, Cu and Ni valence states after nitrate reduction. The decrease of EXAFS peak intensity after nitrate reduction demonstrates that the structure becomes less ordered. For Co, according to the fitting of valence states and energies in Fig. R#3-13, the valence states of Co in RS-20 and Li-RS-16 decreases from 1.94 to 1.67 and 2.39 to 2.14, respectively. In Fig. R#3-12b, no metallic Co is observed by comparing the Co EXAFS results between our catalysts after nitrate reduction and Co foil. When similar analyses are applied to Ni and Cu, the slight reduction can be observed, and no metallic Ni/Cu signal can be found (Fig. R#3-12c-f). In addition, the Cu_2O pre-edge at 8983.7 eV in XANES is not observed after nitrate reduction (Fig. R#3-12e). It indicates that even though Cu_2O and Cu are formed after nitrate reduction, the ratio was very low.

We have added the relevant discussion like above into the revised manuscript (highlighted, following Fig. 3).

Fig. R#3-12. XANES of (a) Co, (c) Ni and (e) Cu K-edges and EXAFS of (b) Co, (d) Cu and (f) Ni K-edges of $\text{Mg}_{0.20}\text{Co}_{0.20}\text{Ni}_{0.20}\text{Cu}_{0.20}\text{Zn}_{0.20}\text{O}$ (RS-20) and $\text{Li}_{0.20}\text{Mg}_{0.16}\text{Co}_{0.16}\text{Ni}_{0.16}\text{Cu}_{0.16}\text{Zn}_{0.16}\text{O}$ (Li-RS-16) before and after nitrate reduction at -0.35 V for 30 min in 1 M KOH + 0.1 M KNO_3 . Standard metals and metal oxides for reference data are attached in the figure. We have added it into the revised manuscript Fig. 3.

Fig. R#3-13. The estimation of Co valence states for the samples from the linear fitting of valence state and energy ($\mu\chi = 0.5$) of CoO and Co₃O₄ XANES. We have added it into the revised Supplementary Fig. 4

We used EELS to probe the electronic states of Co and Ni and Cu at different locations of the bulk and surface before and after nitrate reduction (Fig. R#3-14-17). Statistical results are given in Fig. R#3-18. The change of Co L₃/L₂ intensity ratio can describe the change in Co valence states (*Wang et al., Micron, 2000, 31, 571-580*). Smaller ratios represent higher valence states. After nitrate reduction, Co L₃/L₂ ratios in RS-20 and Li-RS-16 increase at the surface and bulk, with the ratios at the surface bigger than the bulk. It is consistent with XANES results that Co can be reduced after reduction (Fig. 3). The Co valence state trend of RS-20 and Li-RS-16 from EELS measurement is agreement with our XANES results. Moreover, the bigger L₃/L₂ ratio at the surface indicates a bigger reduction degree. For Ni, the intensity between Ni L₃ and L₂ peaks can be used to evaluate the Ni valence state, and a higher intensity after the L₃ peak represents a lower valence state (*Vila' et al., Cell Reports Physical Science, 2020, 1, 100188*). For Cu, the location of the Cu L₃ peak can be used to evaluate the Cu valence state, and the peak location in the higher energy means a lower valence state (*Laffont et al., Micron, 2006, 37, 459-464*). In our EELS results, an obvious reduction of Ni and Cu can be observed, it can be explained through combining with XANES results that the reduction of Ni and Cu is slight (Fig. R#3-15 and 17). We have added the relevant discussion like above into the revised Supplementary Information following Supplementary Fig. 29.

Fig. R#3-14. HAADF-STEM images for EELS measurements of RS-20 prior to and after nitrate reduction at -0.35 V for 30 min in 1 M KOH + 0.1 M KNO₃. We have added it into the revised Supplementary Fig. 25.

Fig. R#3-15. Co, Ni and Cu EELS measurements for the bulk and surface of RS-20 prior to and after nitrate reduction at -0.35 V for 30 min in 1 M KOH + 0.1 M KNO₃: (a) bulk prior to nitrate reduction, (b) surface prior to nitrate reduction, (c) bulk after nitrate reduction and (d) surface after nitrate reduction. All the data are aligned with oxygen peaks in Fig. 2f. We have added it into the revised Supplementary Fig. 26.

Fig. R#3-16. HAADF-STEM images for EELS measurements of Li-RS-16 prior to and after nitrate reduction at -0.35 V for 30 min in 1 M KOH + 0.1 M KNO_3 . We have added it into the revised Supplementary Fig. 27.

Fig. R#3-17. Co, Ni and Cu EELS measurements for the bulk and surface of Li-RS-16 prior to and after nitrate reduction at -0.35 V for 30 min in 1 M KOH + 0.1 M KNO_3 : (a) bulk prior to nitrate reduction, (b) surface prior to nitrate reduction, (c) bulk after nitrate reduction and (d) surface after nitrate reduction. All the data are aligned with oxygen peaks in Fig. 2f. We have added it into the revised Supplementary Fig. 28.

Fig. R#3-18. Co L₃/L₂ ratios from EELS measurements for the bulk and surface of RS-20 and Li-RS-16 prior to and after nitrate reduction at -0.35 V for 30 min in 1 M KOH + 0.1 M KNO₃. We have added it into the revised Supplementary Fig. 29.

In addition, it has been reported that bulk thermochemistry is an excellent descriptor for the catalytic activity trends of oxide surfaces. Bulk thermochemistry and surface adsorption energetics depend similarly on the number of outer electrons of the transition metal in the oxide. This correspondence gives a linear relationship between bulk and surface properties for constructing the volcano-type activity plots (*Calle-Vallejo et al., ACS Catal. 2015, 5, 869–873*).

For the reduction of the oxides, could the electrical conductivity of the oxides be reported? Is it an issue of electrical conductivity why the bulk cannot be reduced? I am unfamiliar with these oxides and their use as electrocatalysts. How much does the pure conductivity change for the different oxides and how would this influence the results?

Response 8: We thank Reviewer #3 for raising this issue. Bérardan et al. reported that $\text{Mg}_{0.20}\text{Co}_{0.20}\text{Ni}_{0.20}\text{Cu}_{0.20}\text{Zn}_{0.20}\text{O}$ has a band gap of ~ 0.8 eV, and the Li introduction decreases the band gap (Bérardan et al. *Phys. Status Solidi RRL*, 2016, 10, 4, 328–333). It means that Li introduction increases the oxide conductivity.

To measure the conductivity of $\text{Mg}_{0.25}\text{Co}_{0.25}\text{Ni}_{0.25}\text{Zn}_{0.25}\text{O}$ (RS-0), $\text{Mg}_{0.20}\text{Co}_{0.20}\text{Ni}_{0.20}\text{Cu}_{0.20}\text{Zn}_{0.20}\text{O}$ (RS-20) and $\text{Li}_{0.20}\text{Mg}_{0.16}\text{Co}_{0.16}\text{Ni}_{0.16}\text{Cu}_{0.16}\text{Zn}_{0.16}\text{O}$ (Li-RS-16), we prepared pellets composed of these oxides without carbon and binder by pressing the raw powders under eight tons followed by heating at 1000 °C for 1 hour with a ramping rate 5 °C min^{-1} (Fig. R#3-19). We have attempted to use the two-electrode method by the multimeter and 4-pin probe method by Loresta-GP MCP-T610 (10 m Ω - 10 M Ω) to obtain the resistivity. However, the resistivities of these three oxides exceed the range of the instrument, which means that these three oxides have poor conductivity.

Fig. R#3-19. The photo of $\text{Mg}_{0.25}\text{Co}_{0.25}\text{Ni}_{0.25}\text{Zn}_{0.25}\text{O}$ (RS-0), $\text{Mg}_{0.20}\text{Co}_{0.20}\text{Ni}_{0.20}\text{Cu}_{0.20}\text{Zn}_{0.20}\text{O}$ (RS-20) and $\text{Li}_{0.20}\text{Mg}_{0.16}\text{Co}_{0.16}\text{Ni}_{0.16}\text{Cu}_{0.16}\text{Zn}_{0.16}\text{O}$ (Li-RS-16) pellets for conductivity measurement.

For the question “Is it an issue of electrical conductivity why the bulk cannot be reduced?”, we agree on that the electric conductivity is indeed an important factor in reducing the oxide. However, in the work, the situation is different. In our previous response, we also observed the metal slight reduction after nitrate reduction by XANES and EELS in **Response 7**. For the electrode for comparing the activity and ammonia yield, carbon was added to mix with these oxides for constructing the conductive network for increase the utilization of those oxides with poor conductivity. The effect from carbon on poorly conductive electrocatalytic materials is investigated and reported (Chung et al., *ACS Catal.* 2020, 10, 4990–4996; Beall et al., *ACS Catal.* 2021, 11, 3094–3114). Hence, under the existence of the

conductive network, we think the conductivity is not the major factor that influence the activity and oxide reduction.

Besides, we can also find some important information on the inconsistency between conductivity and bulk oxide reduction from the previous reports and our results. CuO is a p-type semiconductor with a band gap 1.2 eV in bulk and 1.2-2.1 eV in nanostructured CuO (*Zhang et al., Prog. Mater. Sci., 2014, 60, 208–337*). $\text{Mg}_{0.20}\text{Co}_{0.20}\text{Ni}_{0.20}\text{Cu}_{0.20}\text{Zn}_{0.20}\text{O}$, has a band gap ~ 0.8 V (*Bérardan et al. Physica Status Solidi RRL, 2016, 10, 4, 328–333*). It means that $\text{Mg}_{0.20}\text{Co}_{0.20}\text{Ni}_{0.20}\text{Cu}_{0.20}\text{Zn}_{0.20}\text{O}$ has a better conductivity than CuO. The cation reduction on CuO during nitrate reduction can be observed obviously (*Wang et al., Angew. Chem. Int. Ed. 2020, 59, 5350–5354; Xu et al., Appl. Catal. B: Environ., 2023, 320, 121981*). In contrast, the Cu reduction in $\text{Mg}_{0.20}\text{Co}_{0.20}\text{Ni}_{0.20}\text{Cu}_{0.20}\text{Zn}_{0.20}\text{O}$ is slight. It implies that except for conductivity, under a certain applied potential, other factors such as the oxide composition and crystal structure could affect the oxide reduction.

I cannot comment in detail on the DFT, apart from saying that it is uncertain how the structures used for the calculations match the actual structure under operating condition of the experimental electrocatalyst.

Response 9: We appreciate your valuable comment. Detecting and understanding of catalyst structure in operando conditions at the atomic level remains a big problem in electrocatalysis. In this work, the experimental electrocatalyst was simplified to MgO(111) with Co and Cu-doping in the DFT model, considering the computational efficiency and methodological accuracy. We added additional description to about this issue in the revised manuscript (highlighted) as below.

“DFT Models rationale. Experimental results show that the Mg, Ni, and Zn in HEOs are not active sites but rather play a role in stabilizing HEO structures. In DFT simulations, the Hubbard + U correction (DFT + U) need to be employed to describe strong electron correlation effects in NiO and ZnO, but it is not required in MgO. Therefore, to simplify DFT simulations of (Mg,Ni,Zn)O HEO, we used MgO, taking into account computational efficiency and methodological accuracy. In addition, there are two reasons for the choice of MgO(111): (1) in our experiments, pH is close to 14. It was reported that the hydroxylated (111) surface has a lower surface energy than MgO(100) and MgO(110) (*Geysermans et al., Phys. Chem. Chem. Phys., 2009, 11, 2228–2233*). (2) The MgO(111) surface model (with a specific termination) we used in this work has been well studied in the literature (*Zhang and Tang, J. Phys. Chem. C 2008, 112, 3327-3333*). On the MgO(111) surface, all possible substitutional sites of Cu and Co dopants were calculated (Supplementary Figs. 33-34), including the surface and subsurface, and the most stable structures were then chosen to represent the experimental electrocatalysts.”

General reporting

Reference voltage should be mentioned in all places in the text, as well as electrolyte. The electrolyte plays a huge role on performance and it is not made clear what conditions these are studied in. Is there a change in the pH during operation? If so it should be clearly stated.

Response 10: We thank Reviewer #3 for pointing out these issues. We have supplemented these details in our revised manuscript and Supplementary Information. All the potential is against the reversible hydrogen electrode (RHE). When comparing nitrate reduction and ammonia generation of $\text{Mg}_{0.25}\text{Co}_{0.25}\text{Ni}_{0.25}\text{Zn}_{0.25}\text{O}$, $\text{Mg}_{0.20}\text{Co}_{0.20}\text{Ni}_{0.20}\text{Cu}_{0.20}\text{Zn}_{0.20}\text{O}$, $\text{Li}_{0.20}\text{Mg}_{0.16}\text{Co}_{0.16}\text{Ni}_{0.16}\text{Cu}_{0.16}\text{Zn}_{0.16}\text{O}$, the electrolyte was 1 M KOH and 0.1 M KNO_3 .

We did not observe the pH change during operation. It can be explained as below. For nitrate reduction to ammonia in alkaline, according to the established equation $\text{NO}_3^- + 6\text{H}_2\text{O} + 8\text{e}^- \rightarrow \text{NH}_3 + 9\text{OH}^-$ (*Chen et al., Nat. Nanotechnol., 2022, 17, 759–767*), the formed OH^- can be evaluated by calculating the electrons transferred. More electrons transferred means more OH^- and thus a higher pH value. Thus, we used the largest current density to calculate the change of pH value. We chose the chronoamperometry (CA) data of RS-20 at -0.45 V for 30 min in the revised Supplementary Fig. 11. The integral charge is -491.84 mA s (-0.49184 C), which corresponds to 5.0976 μmol electrons and can be converted to 5.7347 μmol OH^- . In our H-cell setup, the catholyte and anolyte chambers have 18 mL of 1 KOH + 0.1 M KNO_3 , respectively. When considering that no other OH^- exchange from the anolyte chamber, the estimated current $\text{pH} = 14 + \log = 14.000138$. The pH value change can be ignored. Moreover, the OER $8\text{OH}^- \rightarrow 2\text{O}_2 + 4\text{H}_2\text{O} + 8\text{e}^-$ occurs at the anode, and the OH^- will transport to the anode chamber via the anion exchange membrane to keep the OH^- concentration constant between the cathode and anode chambers.

What is the reason for the decrease in FE at high overpotentials in Figure 1h? It is not sufficiently explained or discussed.

Response 11: We thank Reviewer #3 for raising this issue. We supplemented the discussion in the revised manuscript as below.

“The FE order of these oxides is in accordance with their respective ammonia production rate. The FEs of these oxides do not increase monotonically with decreasing applied potential and they peak at -0.35 V and then decrease at -0.45 V. At -0.05 V, RS-0, RS-20 and Li-RS-16 show 33.6%, 72.5% and 51.2% in FE, respectively. At -0.35 V, the FEs of RS-0, RS-20, and Li-RS-16 are 81.8%, 93.4% and 86.7%, respectively. This is due to the competition between ammonia production and the parasitic HER. The obvious HER in the absence of nitrate can be observed, particularly on RS-20. When the applied potential decreases to more negative than the HER thermodynamic potential, the HER is more favored due to its larger charge transfer coefficient for HER than that of *NO hydrogenation. As this disfavors ammonia production, the ammonia FE drops dramatically, particularly observed in those transition metals with a high HER activity, such as Ni. This phenomenon is also reported in other reports (*Carvalho et al., J. Am. Chem. Soc. 2022, 144, 14809-14818; Chen et al., Nat. Nanotechnol., 2022, 17, 759-767; Yang et al., ACS Sustainable Chem. Eng. 2022, 10, 14343-14350*).”

There is insufficient detail in the figure captions. For example, Figure 1 does not include the nitrate concentration, an enormously large factor in nitrate reduction current density.

Response 12: We thank Reviewer #3 for pointing out this issue. We have added sufficient details in the figure caption in the revised version.

In the discussion and results when the flow cell was used, this seems to imply earlier results may have been transport limited? This means that the trends observed may not be valid in the H-cell, because they were transport limited. This needs to be addressed.

Response 13: Thank you for your question. According to Fick's laws, the current density is proportional to the flux of reactants, and the flux is proportional to the concentration gradient along the diffusion layer. Moreover, the concentration gradient is proportional to the concentration difference between the bulk solution and the electrode surface, where the concentrations of the reactants on the electrode surface depend not only the mass transport but also the consumption rate/kinetics. Therefore, although the mass transport in the H-cell is much slower than the flow cell, the nitrate reduction reaction in the H-cell in this work is influenced by both kinetics and mass transport.

One criterion to evaluate our catalysts is to compare the current density at a certain potential. A higher current density represents a higher catalytic activity. When the mass transport influences the current density, the increase of current density will be limited and even the current density reaches a plateau with the increase of overpotential. For our cases in the H-cell, the one affected the most by the mass transport is RS-20, which has the highest current density. For RS-0 and Li-RS-16, which have relatively lower current densities, the influence from mass transport is less dramatic. With the increase of overpotential, no current density plateau was observed and the increase of current density on RS-20 did not slow down compared to RS-0 and Li-RS-16 in Fig. 1e and 1f in the original manuscript. Under the circumstance that the influence from mass transport exists in the H-cell, the trends should be weakened but still be clear. Therefore, the trends observed in the H-cell are reliable.

In the work by Guo et al. titled "Mass Transport Modifies the Interfacial Electrolyte to Influence Electrochemical Nitrate Reduction" published in *ACS Sustainable Chem. Eng.* 2023, 11, 7882-7893, they investigated the influence from mass transport to nitrate reduction. They indicated that at higher flow rates, the equivalent diffusion layer thickness is smaller, and the total current density is higher. The nitrate reduction reaction favors ammonia more under the lowest flow rate than under the highest flow rate.

For the flow cell, could the current for applied case vs. geometric current density also be reported? This would be helpful for context and reference. Also, in the methods I did not quite understand what the gas was for, why is a gas stream needed? I assumed that pure nitrate liquid electrolyte would be fed to the cathode.

Response 14: Thank you for your questions. We have added the LSV curves of RS-20 in 1 M KOH with/without KNO_3 of various concentrations (Fig. R#3-20), as well as the chronoamperometry curves at different applied potentials in 1 M KOH with various concentration KNO_3 in (Fig. R#3-21).

The gas steam is used to purge the gas out of the electrolyte to avoid the possible oxygen reduction reaction, which could sacrifice FE towards ammonia production.

Yes, nitrate electrolyte is also fed to cathode.

Fig. R#3-20. LSV curves of RS-20 in 1 M KOH in the absence of KNO_3 and in the presence of 0.01 M, 0.05 M, 0.1 M and 0.5 M KNO_3 at a scan rate of 10 mV s^{-1} with 85% IR correction in the flow cell. The area of the working electrode was $2 \times 2 \text{ cm}^2$, and the catalyst loading was $0.5 \text{ mg}_{\text{oxide}} \text{ cm}^{-2}$.

We have added it into the revised Supplementary Fig. 30.

Fig. R#3-21. Chronoamperometry curves in the flow cell. RS-20 in (a) 1 M KOH + 0.01 M KNO₃, (b) 1 M KOH + 0.05 M KNO₃, (c) 1 M KOH + 0.1 M KNO₃, and (d) 1 M KOH + 0.1 M KNO₃. The area of the working electrode was $2 \times 2 \text{ cm}^2$, and the catalyst loading was $0.5 \text{ mg}_{\text{oxide}} \text{ cm}^{-2}$. We have added it into the revised Supplementary Fig. 31.

REVIEWERS' COMMENTS

Reviewer #1 (Remarks to the Author):

In the revised manuscript, I think the authors have revised all the points according to the review comments and I think it can be published without further revision.

Reviewer #2 (Remarks to the Author):

The authors provide a suitable response and revision to the reviewers' concerns. Thus, this manuscript can be accepted at its present version.